# Conformal PID Control for Time Series Prediction

**Anastasios N. Angelopoulos**
University of California, Berkeley
angelopoulos@berkeley.edu

**Emmanuel J. Candès**
Stanford University
candes@stanford.edu

**Ryan J. Tibshirani**
University of California, Berkeley
ryantibs@berkeley.edu

## Abstract

We study the problem of uncertainty quantification for time series prediction, with the goal of providing easy-to-use algorithms with formal guarantees. The algorithms we present build upon ideas from conformal prediction and control theory, are able to prospectively model conformal scores in an online setting, and adapt to the presence of systematic errors due to seasonality, trends, and general distribution shifts. Our theory both simplifies and strengthens existing analyses in online conformal prediction. Experiments on 4-week-ahead forecasting of statewide COVID-19 death counts in the U.S. show an improvement in coverage over the ensemble forecaster used in official CDC communications. We also run experiments on predicting electricity demand, market returns, and temperature using autoregressive, Theta, Prophet, and Transformer models. We provide an extendable codebase for testing our methods and for the integration of new algorithms, data sets, and forecasting rules.[1]

## 1 Introduction

Machine learning models run in production systems regularly encounter data distributions that change over time. This can be due to factors such as seasonality and time-of-day, continual updating and re-training of upstream machine learning models, changing user behaviors, and so on. These distribution shifts can degrade a model's predictive performance. They also invalidate standard techniques for uncertainty quantification, such as *conformal prediction* [36, 35].

To address the problem of shifting distributions, we consider the task of prediction in an adversarial online setting, as in [16]. In this problem setting, we observe a (potentially) adversarial time series of deterministic covariates $x_t \in \mathcal{X}$ and responses $y_t \in \mathcal{Y}$, for $t \in \mathbb{N} = \{1, 2, 3, \dots\}$. As in standard conformal prediction, we are free to define any *conformal score function* $s_t : \mathcal{X} \times \mathcal{Y} \to \mathbb{R}$, which we can view as measuring the accuracy of our forecast at time $t$. We will assume with a loss of generality that $s_t$ is negatively oriented (lower values mean greater forecast accuracy). For example, we may use the absolute error $s_t(x, y) = |y - f_t(x)|$, where $f_t$ is a forecaster trained on data up to but not including data at time $t$.

The challenge in the sequential setting is as follows. We seek to invert the score function to construct a *conformal prediction set*,

$$\mathcal{C}_t = \{y \in \mathcal{Y} : s_t(x_t, y) \leq q_t\}, \tag{1}$$

where $q_t$ is an estimated $1 - \alpha$ quantile for the distribution of the score $s_t(x_t, y_t)$ at time $t$. In standard conformal prediction, we would take $q_t$ to be a level $1 - \alpha$ sample quantile (up to a finite-sample correction) of $s_t(x_i, y_i)$, $i < t$; if the data sequence $(x_i, y_i)$, $i \in \mathbb{N}$ were i.i.d. or exchangeable, then this would yield $1 - \alpha$ coverage [35] at each time $t$. However, in the sequential setting, which does not assume exchangeability (or any probabilistic model for the data for that matter), choosing $q_t$ in (1) to yield coverage is a formidable task. In fact, if we are not willing to make any assumptions about the data sequence, then a coverage guarantee at time $t$ would only be possible with trivial methods, which construct prediction intervals of infinite sizes.

---

[1] http://github.com/aangelopoulos/conformal-time-series

37th Conference on Neural Information Processing Systems (NeurIPS 2023).

Therefore, our goal is to achieve *long-run coverage* in time. That is, letting $\mathrm{err}_t = \mathbb{1}\{y_t \notin \mathcal{C}_t\}$, we would like to achieve, for large integers $T$,

$$\frac{1}{T}\sum_{t=1}^{T}\mathrm{err}_t = \alpha + o(1) \tag{2}$$

under few or no assumptions, where $o(1)$ denotes a quantity that tends to zero as $T \to \infty$. We note that (2) is not probabilistic at all, and every theoretical statement we will make in this paper holds deterministically. Furthermore, going beyond (2), we also seek to design flexible strategies to produce the sharpest prediction sets possible, which not only adapt to, but also anticipate distribution shifts.

We call our proposed solution *conformal PID control*. It treats the system for producing prediction sets as a proportional-integral-derivative (PID) controller. In the language of control, the prediction sets take a *control variable*, $q_t$, and then produce a *process variable*, $\mathrm{err}_t$. We seek to anchor $\mathrm{err}_t$ to a *set point*, $\alpha$. To do so, we apply corrections to $q_t$ based on the error of the output, $g_t = \mathrm{err}_t - \alpha$. By reframing the problem in this language, we are able to build algorithms that have more stable coverage while also prospectively adapting to changes in the score sequence, much in the same style as a control system. See the diagram in Figure 1.

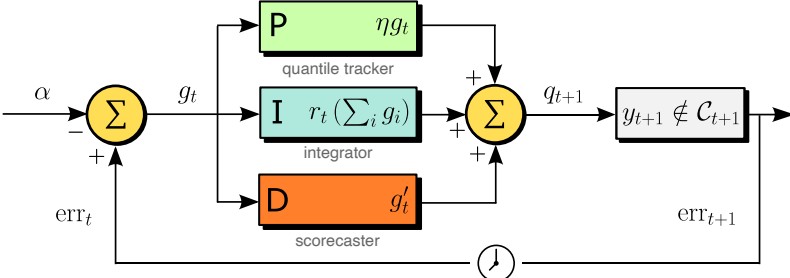

Figure 1: Conformal PID Control, expressed as a block diagram.

## 1.1 Peek at results: methods

Three design principles underlie our methods:

1. *Quantile tracking (P control).* Running online gradient descent on the quantile loss (summed over all past scores) gives rise to a method that we call *quantile tracking*, which achieves long-run coverage (2) under no assumptions except boundedness of the scores. This bound can be unknown. Unlike adaptive conformal inference (ACI) [16], quantile tracking does not return infinite sets after a sequence of miscoverage events. This can be seen as equivalent to proportional (P) control.

2. *Error integration (I control).* By incorporating the running sum $\sum_{i=1}^{t}(\mathrm{err}_i - \alpha)$ of the coverage errors into the online quantile updates, we can further stabilize the coverage. This *error integration* scheme achieves long-run coverage (2) under no assumptions whatsoever on the scores (they can be unbounded). This can be seen as equivalent to integral (I) control.

3. *Scorecasting (D control).* To account for systematic trends in the scores—this may be due to aspects of the data distribution, fixed or changing, which are not captured by the initial forecaster—we train a second model, a *scorecaster*, to predict the quantile of the next score. While quantile tracking and error integration are merely reactive, scorecasting is forward-looking. It can residualize out systematic trends in the errors and lead to practical advantages in terms of coverage and efficiency (set sizes). This can be seen as equivalent to derivative (D) control. Traditional control theory would suggest using a linear approximation $g'_t = g_t - g_{t-1}$, but in our problem, we will typically choose more advanced scorecasting algorithms that go well beyond simple difference schemes.

These three modules combine to make our final iteration, the *conformal PID controller*:

$$q_{t+1} = \underbrace{\eta g_t}_{P} + \underbrace{r_t\left(\sum_{i=1}^{t} g_t\right)}_{I} + \underbrace{g'_t}_{D}. \tag{3}$$

In traditional PID control, one would take $r_t(x)$ to be a linear function of $x$. Here, we allow for nonlinearity and take $r_t$ to be a *saturation function* obeying

$$x \geq c \cdot h(t) \implies r_t(x) \geq b, \quad \text{and} \quad x \leq -c \cdot h(t) \implies r_t(x) \leq -b, \tag{4}$$

for constants $b, c > 0$, and a sublinear, nonnegative, nondecreasing function $h$—we call a function $h$ satisfying these conditions *admissible*. An example is the *tangent integrator* $r_t(x) = K_I \tan(x \log(t)/(t C_{\text{sat}}))$, where we set $\tan(x) = \text{sign}(x) \cdot \infty$ for $x \notin [-\pi/2, \pi/2]$, and $C_{\text{sat}}, K_I > 0$ are constants. The choice of integrator $r_t$ is a design decision for the user, as is the choice of scorecaster $g'_t$.

We find it convenient to reparametrize (3), to produce a sequence of quantile estimates $q_t$, $t \in \mathbb{N}$ used in the prediction sets (1), as follows:

$$\text{let } \hat{q}_{t+1} \text{ be any function of the past: } x_i, y_i, q_i, \text{ for } i \leq t,$$

$$\text{then update } q_{t+1} = \hat{q}_{t+1} + r_t\left( \sum_{i=1}^{t} (\text{err}_i - \alpha) \right). \tag{5}$$

Taking $\hat{q}_{t+1} = \eta g_t + g'_t$ recovers (3), but we find it generally useful to instead consider the formulation in (5), which will be our main focus in the exposition henceforth. Now we view $\hat{q}_{t+1}$ as the scorecaster, which directly predicts $q_{t+1}$ using past data. A main result of this paper, whose proof is given in Appendix A, is that the conformal PID controller (5) yields long-run coverage for any choice of integrator $r_t$ that satisfies the appropriate saturation condition, and any scorecaster $\hat{q}_{t+1}$.

**Theorem 1.** *Let $\{\hat{q}_t\}_{t \in \mathbb{N}}$ be any sequence of numbers in $[-b/2, b/2]$ and let $\{s_t\}_{t \in \mathbb{N}}$ be any sequence of score functions with outputs in $[-b/2, b/2]$. Here $b > 0$, and may be infinite. Assume that $r_t$ satisfies (4), for an admissible function $h$. Then the iterations in (5) achieve long-run coverage (2).*

To emphasize, this result holds deterministically, with no probabilistic model on the data $(x_t, y_t)$, $t \in \mathbb{N}$. (Thus in the case that the sequence is random, the result holds for all realizations of the random variables.) As we will soon see, this theorem can be seen as a generalization of existing results in the online conformal literature.

## 1.2 Peek at results: experiments

**COVID-19 death forecasting.** To demonstrate conformal PID in practice, we consider 4-week-ahead forecasting of COVID-19 deaths in California, from late 2020 through late 2022. The base forecaster $f_t$ that we use is the ensemble model from the COVID-19 Forecast Hub, which is the model used for official CDC communications on COVID-19 forecasting [10, 29]. In this forecasting problem, at each time $t$ we actually seek to predict the observed death count $y_{t+4}$ at time $t + 4$.

Figure 2 shows the central 80% prediction sets from the Forecast Hub ensemble model on the left panel, and those from our conformal PID method on the right. We use a quantile conformal score function, as in conformalized quantile regression [30], applied asymmetrically (i.e., separately) to the lower and upper quantile levels). We use the tan integrator, with constants chosen heuristically (as described in Appendix C), and an $\ell_1$-regularized quantile regression as the scorecaster—in particular, the scorecasting model at time $t$ predicts the quantile of the score at time $t + 4$ based on all previous forecasts, cases, and deaths, from *all 50 US states*. The main takeaway is that conformal PID control is able to correct for consistent underprediction of deaths in the winter wave of late 2020/early 2021. We can see from the figure that the original ensemble fails to cover 8 times in a stretch of 10 weeks, resulting in a coverage of 20%; meanwhile, conformal PID only fails to cover 3 times during this stretch, restoring the coverage to 70% (recall the nominal level is 80%).

How is this possible? The ensemble is mainly comprised of constituent forecasters that ignore geographic dependencies between states [11] for the sake of simplicity or computational tractability. But COVID infections and deaths exhibit strong spatiotemporal dependencies, and most US states experienced the winter wave of late 2020/early 2021 at similar points in time. The scorecaster is thus able to learn from the mistakes made on other US states in order to prospectively adjust the ensemble's forecasts for the state of California. Similar improvements can be seen for other states, and we include experiments for New York and Texas as examples in Appendix F, which also gives more details on the scorecaster and the results.

**Electricity demand forecasting.** Next we consider a data set on electricity demand forecasting in New South Wales [18], which includes half-hourly data from May 7, 1996 to December 5, 1998.

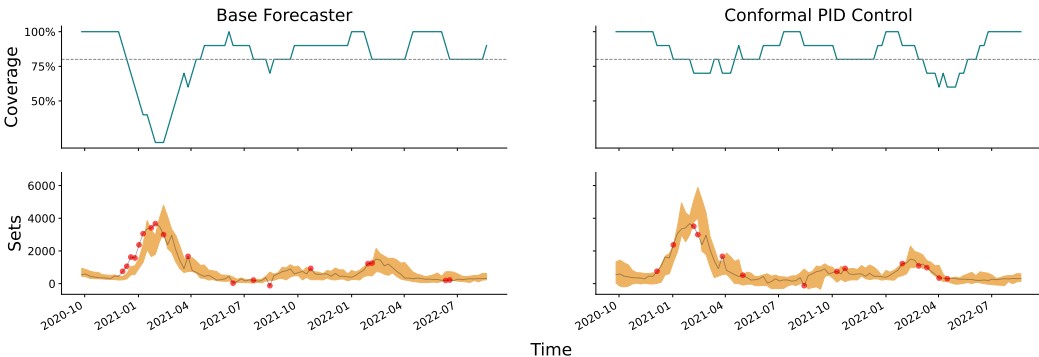

Figure 2: Results for 4-week ahead COVID-19 death forecasting in California. The left column shows the COVID-19 Forecast Hub ensemble model, and the right column shows conformal PID control using the tan integrator, and a scorecaster given by $\ell_1$-penalized quantile regression on all past forecasts, cases, and deaths from all 50 states. The top row plots coverage, averaged over a trailing window of 10 weeks. The nominal coverage level is $1 - \alpha = 0.8$ and marked by a gray dotted line. The bottom row plots the prediction sets in gold along with the ground-truth times series (death counts). Miscoverage events are indicated by red dots. Summary statistics such as the coverage and average set size are available in Table 1.

For the base forecaster we use a Transformer model [34] as implemented in `darts` [19]. This is only re-trained daily, to predict the entire day's demand in one batch; this is a standard approach with Transformer models due to their high computational cost. For the conformal score, we use the asymmetric (signed) residual score. We use the tan integrator as before, and we use a lightweight Theta model [2], re-trained at every time point (half-hour), as the scorecaster.

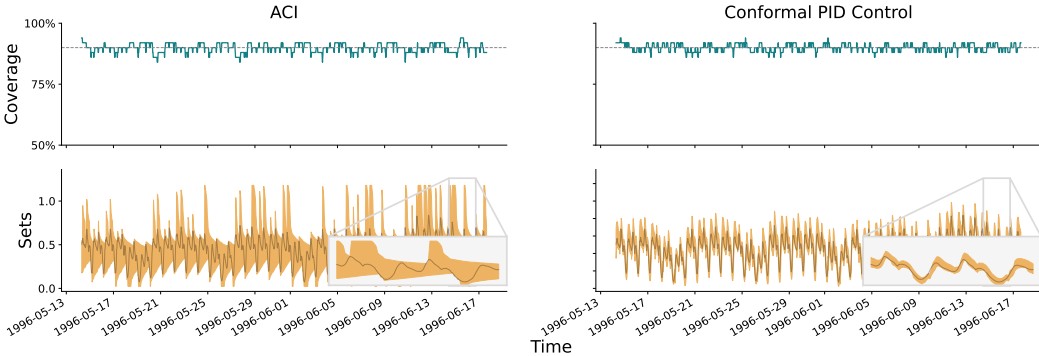

Figure 3: Results for electricity demand forecasting. The left column shows adaptive conformal inference (ACI), and the right column shows conformal PID control. The base forecaster is a Transformer model, and we use a tan integrator and a Theta scorecaster. The format of the figure follows that of Figure 2, except the nominal coverage is now $1 - \alpha = 0.9$, and the coverage is averaged over a trailing window of 50 points (we also omit the red dots which mark miscoverage events). Summary statistics are available in Table 2.

The results are shown in the right panel of Figure 3, where adaptive conformal inference (ACI) [16] is also compared in the left panel. In short, conformal PID control is able to anticipate intraday variations in the scores, and produces sets that "hug" the ground truth sequence tightly; it achieves tight coverage without generating excessively large or infinite sets. The main reason why this is improved is that the scorecaster has a seasonality component built into its prediction model; in general, large improvements such as the one exhibited in Figure 3 should only be expected when the base forecaster is imperfect, as is the case here.

## 1.3 Related work

The adversarial online view of conformal prediction was pioneered by [16] in the same paper that first introduced ACI. Since then, there has been significant work towards improving ACI, primarily by setting the learning rate adaptively [17, 40, 7], and incorporating ideas from multicalibration to improve conditional coverage [5]. It is worth noting that [7] also makes the observation that the ACI iteration can be generalized to track the quantile of the score sequence, although their focus is on adaptive regret guarantees. Because the topic of adaptive learning rates for ACI and related algorithms has already been investigated heavily, we do not consider it in the current paper. Any such method, such as those of [17, 7] should work well in conjunction with our proposed algorithms.

A related but distinct line of work surrounds online *calibration* in the adversarial sequence model, which dates back to [14, 15], and connects in interesting ways to both game theory and online learning. We will not attempt to provide a comprehensive review of this rich and sizeable literature, but simply highlight [25, 24, 23] as a few interesting examples of recent work.

Lastly, outside the online setting, we note that several researchers have been interested in generalizing conformal prediction beyond the i.i.d. (or exchangeable) data setting: this includes [33, 28, 26, 12, 8], and for time series prediction, in particular, [9, 31, 38, 39, 3]. The focus of all of these papers is quite different, and they all rely on probabilistic assumptions on the data sequence to achieve validity; we include further discussion in Appendix B.

## 2 Methods

We describe the main components of our proposal one at a time, beginning with the quantile tracker.

### 2.1 Quantile tracking

The starting point for quantile tracking is to consider the following optimization problem:

$$\underset{q \in \mathbb{R}}{\text{minimize}} \sum_{t=1}^{T} \rho_{1-\alpha}(s_t - q), \tag{6}$$

for large $T$, where we abbreviate $s_t = s_t(x_t, y_t)$ for the score of the test point, and $\rho_{1-\alpha}$ denotes the quantile loss at the level $1 - \alpha$, i.e., $\rho_\tau(z) = \tau|z|$ for $z > 0$ and $(1 - \tau)|z|$ for $z \le 0$. The latter is the standard loss used in quantile regression [22, 21]. Problem (6) is thus a simple convex (linear) program that tracks the $1 - \alpha$ quantile of the score sequence $s_t$, $t \in N$. To see this, recall that for a continuously distributed random variable $Z$, the expected loss $\mathbb{E}[\rho_{1-\alpha}(Z - q)]$ is uniquely minimized (over $q \in \mathbb{R}$) at the level $1 - \alpha$ quantile of the distribution of $Z$.

In the sequential setting, where we receive one score $s_t$ at a time, a natural and simple approach is to apply *online gradient descent* to (6), with a constant learning rate $\eta > 0$. This results in the update:[2]

$$q_{t+1} = q_t + \eta \nabla \rho_{1-\alpha}(s_t - q_t)$$
$$= q_t + \eta(\text{err}_t - \alpha), \tag{7}$$

where the second line follows as $\nabla \rho_{1-\alpha}(s_t - q_t) = 1 - \alpha$ if $s_t > q_t \iff \text{err}_t = 1$, and $\nabla \rho_{1-\alpha}(s_t - q_t) = -\alpha$ if $s_t \le q_t \iff \text{err}_t = 0$. Note that the update in (7) is highly intuitive: if we miscovered (committed an error) at the last iteration then we increase the quantile, whereas if we covered (did not commit an error) then we decrease the quantile.

Even though it is extremely simple, the quantile tracking iteration (7) can achieve long-run coverage own its own, provided the scores are bounded.

**Proposition 1.** *Let $\{s_t\}_{t \in \mathbb{N}}$ be any sequence of numbers in $[-b, b]$, for $0 < b < \infty$. Then the quantile tracking iteration (7) satisfies*

$$\left| \frac{1}{T} \sum_{t=1}^{T} (\text{err}_t - \alpha) \right| \le \frac{b + \eta}{\eta T},$$

*for any learning rate $\eta > 0$ and $T \ge 1$. In particular, (7) yields long-run coverage as in (2).*

---

[2]Technically, this is the online subgradient method; in a slight abuse of notation, we write $\nabla \rho_{1-\alpha}(0)$ to denote a subgradient of $\rho_{1-\alpha}$ at 0, which can take on any value in $[-\alpha, 1 - \alpha]$.

A few remarks are in order. First, although Proposition 1 assumes boundedness of the scores, we do not need to know this bound in order to run (7) and obtain long-run coverage. As long as the scores lie in $[-b, b]$ for any finite $b$, the guarantee goes through—clearly, the quantile tracker proceeds agnostically and performs the same updates in any case. Notably, the adaptive conformal inference algorithm can be expressed as a special case of the quantile tracker; see Appendix B.1 for details.

Second, for the learning rate, in practice we typically set $\eta$ heuristically, as some fraction of the highest score over a trailing window $\hat{B}_t = \max\{s_{t-\Delta+1}, \dots, s_t\}$. On this scale, setting $\eta = 0.1\hat{B}_t$ usually gives good results, and we use it in all experiments unless specified otherwise (we also set the window length $\Delta$ to be the same as the length of the burn-in period for training the initial base forecaster and scorecaster).[3] Extremely high learning rates result in volatile sets, while very low ones may fail to keep up with rapid changes in the score distribution.

## 2.2 Error integration

Error integration is a generalization of quantile tracking that follows the iteration:

$$q_{t+1} = r_t\left(\sum_{i=1}^{t}(\text{err}_i - \alpha)\right),\tag{8}$$

where $r_t$ is a saturation function that satisfies (4) for an admissible function $h$; recall that we use admissible to mean nonnegative, nondecreasing, and sublinear. As we saw in (13), the quantile tracker uses a *constant* threshold function $h$, whereas $h$ is now permitted to grow, as long as it grows sublinearly, i.e., $h(t)/t \to 0$ as $t \to \infty$. A non-constant threshold function $h$ can be desirable because it means that $r_t$ will "saturate" (will hit the conditions on the right-hand sides in (4)) less often, so corrections for coverage error will occur less often, and in this sense, a greater degree of coverage error can be tolerated along the sequence.

The next proposition, in particular its proof, makes the role of $h$ precise. Importantly, Proposition 2 suffices to prove Theorem 1.

**Proposition 2.** *Let $\{s_t\}_{t \in \mathbb{N}}$ be any sequence of numbers in $[-b, b]$, where $b > 0$, and may be infinite. Assume that $r_t$ satisfies* (4)*, for an admissible function $h$. Then the error integration iteration* (8) *satisfies*

$$\left|\frac{1}{T}\sum_{t=1}^{T}(\text{err}_t - \alpha)\right| \leq \frac{c \cdot h(T) + 1}{T},\tag{9}$$

*for any $T \geq 1$, where $c$ is the constant in* (4)*. In particular, this means* (8) *yields long-run coverage* (2)*.*

The choice of saturation function essentially corresponds to a choice of adaptive learning rate; see Appendix D for details.

## 2.3 Scorecasting

The final piece to discuss is scorecasting. A scorecaster attempts to forecast $q_{t+1}$ directly, taking advantage of any leftover signal that is not captured by the base forecaster. This is the role played by $\hat{q}_{t+1}$ in (5). Scorecasting may be particularly useful when it is difficult to modify or re-train the base forecaster. This can occur when the base forecaster is computationally costly to train (e.g., as in a Transformer model); or it can occur in complex operational prediction pipelines where frequently updating a forecasting implementation is infeasible. Another scenario where scorecasting may be useful is one in which the forecaster and scorecaster have access to different levels of data. For example, if a public health agency collects epidemic forecasts from external groups, and forms an ensemble forecast, then the agency may have access to finer-grained data that it can use to recalibrate the ensemble's prediction sets (compared to the level of data granularity granted to the forecasters originally).

This motivates the need for scorecasting as a modular layer that "sits on top" of the base forecaster and residualizes out systematic errors in the score distribution. This intuition is made more precise

---

[3]Technically, this learning rate is not fixed, so Proposition 1 does not directly apply. However, we can view it as a special case of error integration and an application of Proposition 2 thus provides the relevant coverage guarantee.

by recalling, as described above (following Proposition 2), that scorecasting combined with error integration as in (5) is just a reparameterization of error integration (8), where $q'_t = q_t - \hat{q}_t$ and $s'_t = s_t - \hat{q}_t$ are the new quantile and new score, respectively. A well-executed scorecaster could reduce the variability in the scores and make them more exchangeable, resulting in more stable coverage and tighter prediction sets, as seen in Figure 3. On the other hand, an aggressive scorecaster with little or no signal can actually hurt by adding variance to the new score sequence $s'_t$, which could result in more volatile coverage and larger sets.

There is no limit to what we can choose for the scorecasting model. We might like to use a model that can simultaneously incorporate seasonality, trends, and exogenous covariates. Common choices would be SARIMA (seasonal autoregressive integrated moving average) and ETS (error-trend-seasonality) models, but there are many other available methods, such as the Theta model [2], Prophet model [32], and neural network forecasters; see [20] for a review.

## 2.4 Putting it all together

Briefly, we revisit the PID perspective, to recap how quantile tracking, error integration, and scorecasting fit in and work in combination. It helps to return to (3), which we copy again here:

$$q_{t+1} = g'_t + \eta(\text{err}_t - \alpha) + r_t\left(\sum_{i=1}^{t}(\text{err}_i - \alpha)\right). \tag{10}$$

Quantile tracking is precisely given by taking $g'_t = q_t$ and $r_t = 0$. This can be seen as equivalent to P control: subtract $q_t$ from both sides in (10) and treat the increment $u_{t+1} = q_{t+1} - q_t$ as the process variable; then in this modified system, quantile tracking is exactly P control. For this reason, we use "conformal P control" to refer to the quantile tracker in the experiments that follow. Similarly, we use "conformal PI control" to refer to the choice $g'_t = q_t$, and $r_t \neq 0$ as a generic integrator (for us, tan is the default). Lastly, "conformal PID control" refers to letting $g'_t$ be a generic scorecaster, and $r_t \neq 0$ be a generic integrator.

# 3 Experiments

In addition to the statewide COVID-19 death forecasting experiment described in the introduction, we run experiments on all combinations of the following data sets and forecasters.

Data sets:

- Electricity demand [18]
- Return (log price) of Amazon, Google, and Microsoft stock [27]
- Temperature in Delhi [37]

Forecasters (all via `darts` [19]) :

- Autoregressive (AR) model with 3 lags
- Theta model with $\theta = 2$ [2]
- Prophet model [32]
- Transformer model [34]

In all cases except for the COVID-19 forecasting data set, we: re-train the base forecaster at each time point; construct prediction sets using the asymmetric (signed) residual score; and use a Theta model for the scorecaster. For the COVID-19 forecasting setting, we: use the given ensemble model as the base forecaster (no training at all); construct prediction sets using the asymmetric quantile score; and use an $\ell_1$-penalized quantile regression as the scorecaster, fit on features derived from previous forecasts, cases, and deaths, as described in the introduction. And lastly, in all cases, we use a tan function for the integrator with constants chosen heuristically, as described in Appendx C.

The results that we choose to show in the subsections below are meant to illustrate key conceptual points (differences between the methods). Additional results are presented in Appendix G. Our GitHub repository, `https://github.com/aangelopoulos/conformal-time-series`, provides the full suite of evaluations.

## 3.1 ACI versus quantile tracking

We forecast the daily Amazon (AMZN) opening stock price from 2006–2014. We do this in log-space (hence predicting the return of the stock). Figure 4 compares ACI and the quantile tracker, each

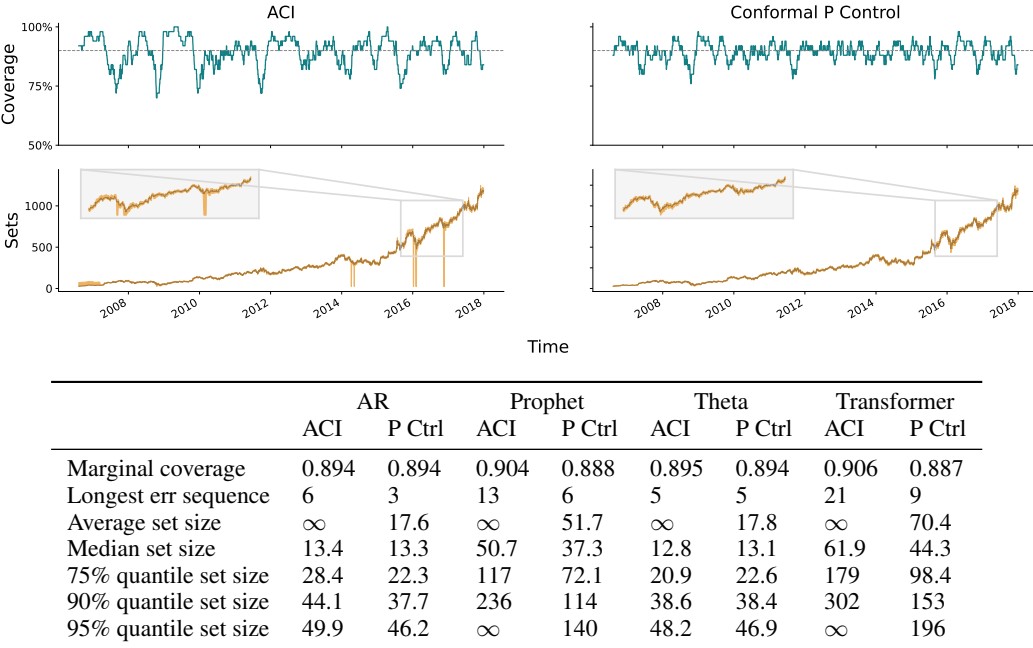

Figure 4: Results for forecasting Amazon stock return, comparing ACI and quantile tracking (P control). The plots show AR as the base forecaster; the table summarizes the results of all four base forecasters. We use the default learning rates for both ACI and quantile tracking: $\eta = 0.005$ and $\eta = 0.1\hat{B}_t$, respectively.

|  | AR | | Prophet | | Theta | | Transformer | |
|---|---|---|---|---|---|---|---|---|
|  | ACI | P Ctrl | ACI | P Ctrl | ACI | P Ctrl | ACI | P Ctrl |
| Marginal coverage | 0.894 | 0.894 | 0.904 | 0.888 | 0.895 | 0.894 | 0.906 | 0.887 |
| Longest err sequence | 6 | 3 | 13 | 6 | 5 | 5 | 21 | 9 |
| Average set size | $\infty$ | 17.6 | $\infty$ | 51.7 | $\infty$ | 17.8 | $\infty$ | 70.4 |
| Median set size | 13.4 | 13.3 | 50.7 | 37.3 | 12.8 | 13.1 | 61.9 | 44.3 |
| 75% quantile set size | 28.4 | 22.3 | 117 | 72.1 | 20.9 | 22.6 | 179 | 98.4 |
| 90% quantile set size | 44.1 | 37.7 | 236 | 114 | 38.6 | 38.4 | 302 | 153 |
| 95% quantile set size | 49.9 | 46.2 | $\infty$ | 140 | 48.2 | 46.9 | $\infty$ | 196 |

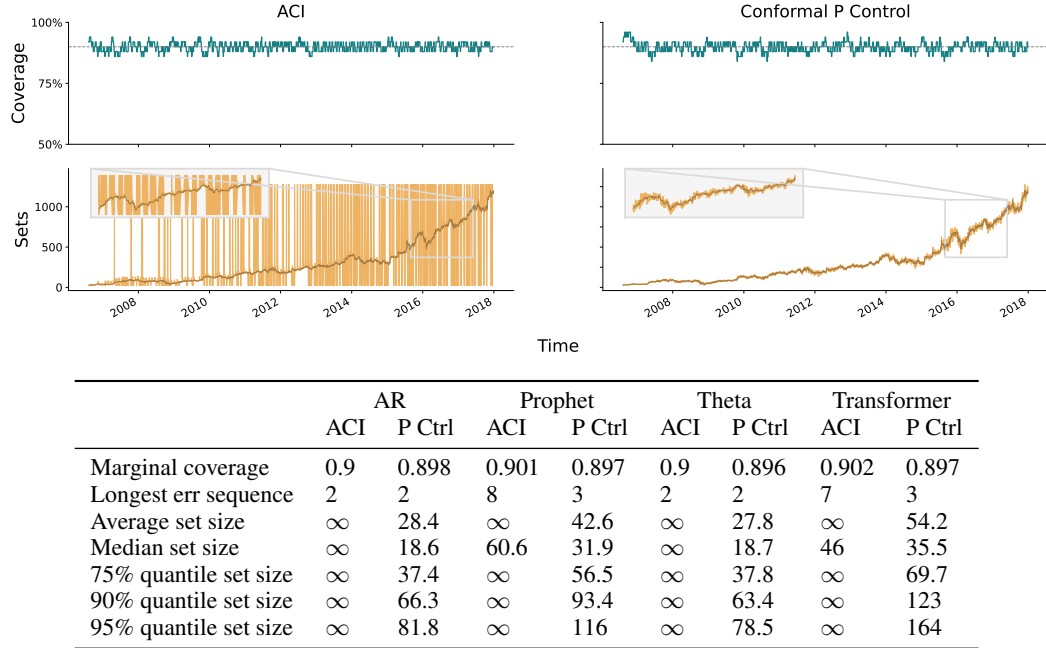

Figure 5: As in Figure 4, but with larger learning rates for ACI and quantile tracking: $\eta = 0.1$ and $\eta = 0.5\hat{B}_t$, respectively.

|  | AR | | Prophet | | Theta | | Transformer | |
|---|---|---|---|---|---|---|---|---|
|  | ACI | P Ctrl | ACI | P Ctrl | ACI | P Ctrl | ACI | P Ctrl |
| Marginal coverage | 0.9 | 0.898 | 0.901 | 0.897 | 0.9 | 0.896 | 0.902 | 0.897 |
| Longest err sequence | 2 | 2 | 8 | 3 | 2 | 2 | 7 | 3 |
| Average set size | $\infty$ | 28.4 | $\infty$ | 42.6 | $\infty$ | 27.8 | $\infty$ | 54.2 |
| Median set size | $\infty$ | 18.6 | 60.6 | 31.9 | $\infty$ | 18.7 | 46 | 35.5 |
| 75% quantile set size | $\infty$ | 37.4 | $\infty$ | 56.5 | $\infty$ | 37.8 | $\infty$ | 69.7 |
| 90% quantile set size | $\infty$ | 66.3 | $\infty$ | 93.4 | $\infty$ | 63.4 | $\infty$ | 123 |
| 95% quantile set size | $\infty$ | 81.8 | $\infty$ | 116 | $\infty$ | 78.5 | $\infty$ | 164 |

with its default learning rate: $\eta = 0.005$ for ACI, and $\eta = 0.1\hat{B}_t$ for quantile tracking. We see that the coverage from each method is decent, but oscillates nontrivially around the nominal level of $1 - \alpha = 0.9$ (with ACI generally having larger oscillations). Figure 5 thus increases the learning rate for each method: $\eta = 0.1$ for ACI, and $\eta = 0.5\hat{B}_t$ for the quantile tracker. We now see that both deliver very tight coverage. However, ACI does so by frequently returning infinite sets; meanwhile, the corrections to the sets made by the quantile tracker are nowhere near as aggressive.

As a final comparison, in Appendix E, we modify ACI to clip the sets in a way that disallows them from ever being infinite. This heuristic may be used by practitioners that want to guard against infinite sets, but it no longer has a validity guarantee for bounded or unbounded scores. The results in the appendix indicate that the quantile tracker has similar coverage to this procedure, and usually with smaller sets.

## 3.2 The effect of integration

Next we forecast the daily Google (GOOGL) opening stock price from 2006–2014 (again done in log-space). Figure 6 compares the quantile tracker without and without an additional integrator component (P control versus PI control). We purposely choose a very small learning rate, $\eta = 0.01\hat{B}_t$, in order to show how the integrator can stabilize coverage, which it does nicely for most of the time series. The coverage of PI control begins to oscillate more towards the end of the sequence, which we attribute at least in part to the fact that the integrator measures coverage errors accumulated over *all time*—and by the end of a long sequence, the marginal coverage can still be close to $1 - \alpha$ even if the local coverage deviates more wildly. This can be addressed by using a local version of the integrator, an idea we return to in the discussion.

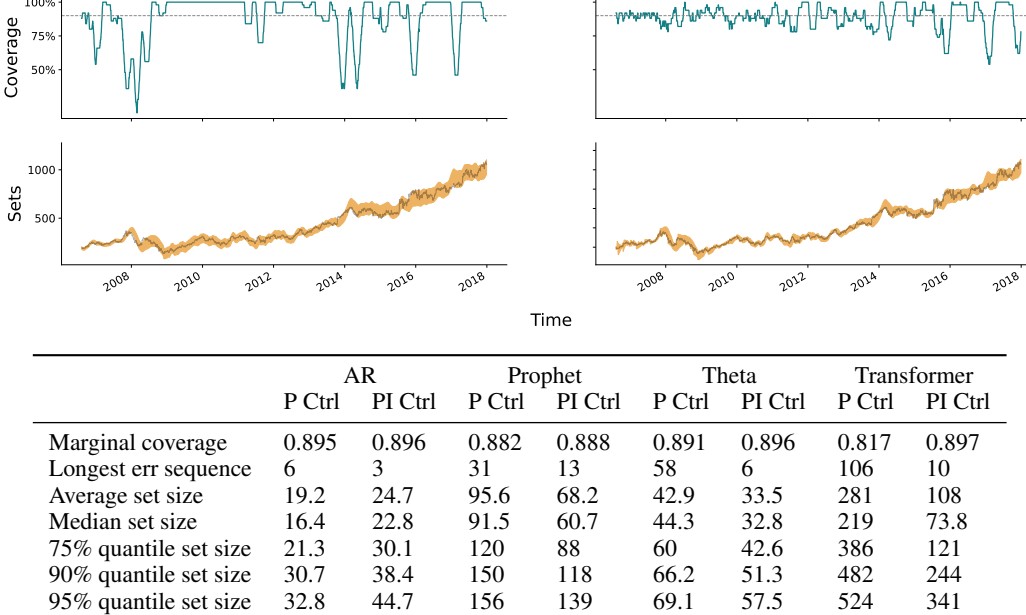

| | AR | | Prophet | | Theta | | Transformer | |
|---|---|---|---|---|---|---|---|---|
| | P Ctrl | PI Ctrl | P Ctrl | PI Ctrl | P Ctrl | PI Ctrl | P Ctrl | PI Ctrl |
| Marginal coverage | 0.895 | 0.896 | 0.882 | 0.888 | 0.891 | 0.896 | 0.817 | 0.897 |
| Longest err sequence | 6 | 3 | 31 | 13 | 58 | 6 | 106 | 10 |
| Average set size | 19.2 | 24.7 | 95.6 | 68.2 | 42.9 | 33.5 | 281 | 108 |
| Median set size | 16.4 | 22.8 | 91.5 | 60.7 | 44.3 | 32.8 | 219 | 73.8 |
| 75% quantile set size | 21.3 | 30.1 | 120 | 88 | 60 | 42.6 | 386 | 121 |
| 90% quantile set size | 30.7 | 38.4 | 150 | 118 | 66.2 | 51.3 | 482 | 244 |
| 95% quantile set size | 32.8 | 44.7 | 156 | 139 | 69.1 | 57.5 | 524 | 341 |

Figure 6: Results for forecasting Google stock return, comparing quantile tracking with and without the integrator (P control versus PI control). The plots show Prophet as the base forecaster; the table summarizes the results of all four base forecasters. We purposely use a very small learning rate, $\eta = 0.01\hat{B}_t$, in order to show how the integrator can stabilize coverage.

## 3.3 The effect of scorecasting

Figures 2 and 3 already showcase examples in which scorecasting offers significant improvement in coverage and set sizes. Recall that these were settings in which the base forecaster produces errors (scores) that have predictable trends. Further examples in the COVID-19 forecasting setting, which display similar benefits to scorecasting, are given in Appendix F.

We emphasize that it is not always the case that scorecasting will help. In some settings, scorecasting may introduce enough variance into the new score sequence that the coverage or sets will degrade in stability. (For example, this will happen if we run a highly complex scorecaster on a sequence of i.i.d. scores, where there are no trends whatsoever.) In practice, scorecasters should be designed with care, just as one would design a base forecaster; it is unlikely that using "out of the box" techniques for scorecasting will be robust enough, especially in high-stakes problems. Appendix G provides examples in which scorecasting, run across all settings using a generic Theta model, can hurt (for example, it adds noticeable variance to the coverage and sets in some instances within the Amazon data setting).

## 4 Discussion and extensions

**Discussion.** Our work presents a framework for constructing prediction sets in time series that is analogous (and indeed formally equivalent) to PID control, consisting of quantile tracking (P control), which is simply online gradient descent applied to the quantile loss; error integration (I control) to stabilize coverage; and scorecasting (D control) to remove systematic trends in the scores (errors made by the base forecaster).

We found that the combination of quantile tracking and integration consistently yields robust and favorable performance in our experiments. Scorecasting provides additional benefits if there are trends left in scores that are predictable (and the scorecaster is well-designed), as is the case in some of our examples. Otherwise, scorecasting may add variability and make the coverage and prediction sets more volatile. Overall, designing the scorecaster (which includes the choice to even use one at all) is an important modeling step, just like the design of the base forecaster.

It is worth emphasizing that, with the exception of the COVID-19 forecasting example, our experiments are intended to be illustrative, and we did not look to use state-of-the-art forecasters, or include any and all possibly relevant features for prediction. Further, while we found that using heuristics to set constants (such as the learning rate $\eta$, and constants $C_{\text{sat}}$, $K_{\text{I}}$ for the tan integrator) worked decently well, we believe that more rigorous techniques, along the lines of [17, 7], can be used to tune these adaptively in an online fashion.

**Extensions.** We now present an extension of our analysis to conformal risk control [1, 6, 13]. In this setting, we are given a sequence of loss functions $L_t : 2^{\mathcal{Y}} \times \mathcal{Y} \to [0, 1]$ satisfying $L_t(\mathcal{Y}, y) = 0$ for all $y$, and $L_t(\emptyset, y) = 1$ for all $y$. The goal is to bound the deviation of the average risk $\frac{1}{T}\sum_{t=1}^{T} L_t(\mathcal{C}_t, y_t)$ from $\alpha$. We state a result for the integrator below, and give its proof in Appendix A.

**Proposition 3.** *Consider the iteration $q_{t+1} = r_t(\sum_{i=1}^{t}(L_i(\mathcal{C}_i, y_i) - \alpha))$, with $L_t$ as above. Assume that $r_t$ satisfies (4), for an admissible function $h$. Also assume that $C_t(\mathcal{C}_t, y_t) = \emptyset$ if $q_t \leq -b$ and $\mathcal{Y}$ if $q_t \geq b$, where $b > 0$, and may be infinite. Then for all $T \geq 1$,*

$$\left| \frac{1}{T}\sum_{t=1}^{T}(L_t(\mathcal{C}_t, y_t) - \alpha) \right| \leq \frac{c \cdot h(T) + 1}{T}. \tag{11}$$

*for any $T \geq 1$, where $c$ is the constant in (4).*

We briefly conclude by mentioning that we believe many other extensions are possible, especially with respect to the integrator. Broadly, we can choose to integrate in a kernel-weighted fashion,

$$r_t\left( \sum_{i=1}^{t}(\text{err}_i - \alpha) \cdot K\big((i, x_i, y_i), (t, x_t, y_t)\big) \right). \tag{12}$$

As a special case, the kernel could simply assign weight 1 if $t - i \leq w$, and weight 0 otherwise, which would result in an integrator that aggregates coverage over a trailing window of length $w$. This can help consistently sustain better local coverage, for long sequences. As another special case, the kernel could assign a weight based on whether $x_i$ and $x_t$ lie in the same bin in some pre-defined binning of $\mathcal{X}$ space, which may be useful for problems with group structure (where we want group-wise coverage). Various other choices and forms of kernels are possible, and it would be interesting to consider adding together a number of such choices (12) in combination, in a multi-resolution flavor, for the ultimate quantile update.

**Acknowledgments**

We would like to thank Tijana Zrnić, Amit Kohli, and Jordan Lekeufack for their valuable feedback. This work was supported by the National Science Foundation (NSF) Graduate Research Fellowship Program under grant no. 2146752, and the Office of Naval Research (ONR) Multi-University Research Initiative (MURI) Program under grant no. N00014-20-1-2787.

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

## A Proofs

We begin with the proof of Proposition 1. The proof is very simple, and we derive it as a corollary of Proposition 2, given next, because the proof reveals something perhaps unforeseen about the quantile tracker: it acts as an error integrator, despite only adjusting the quantile based on the most recent time step.

*Proof of Proposition 1.* Without a loss of generality, set $q_1 = 0$. Unraveling the iteration (7) yields

$$q_{t+1} = \eta \sum_{i=1}^{t} (\text{err}_i - \alpha). \tag{13}$$

For $r_t(x) = \eta x$, $h(t) = b$, we see (4) holds with $c = 1/\eta$. Proposition 2 now applies. □

*Proof of Proposition 2.* Abbreviate $E_T = \sum_{t=1}^{T} (\text{err}_t - \alpha)$. We will prove one side of the absolute inequality in (9), namely, $E_T \leq c \cdot h(T) + 1$, and the other side follows similarly. We use induction. The base case, for $T = 1$, holds trivially. Now assume the result is true up to $T - 1$. We divide the argument into two cases: either $c \cdot h(T-1) < E_{T-1} \leq c \cdot h(T-1) + 1$ or $E_{T-1} \leq c \cdot h(T-1)$. In the first case, note that that (4) implies $q_T = r_t(E_{T-1}) \geq b$ and therefore $s_T \leq q_T$ and $\text{err}_t = 0$. This means that

$$E_T = E_{T-1} - \alpha \leq c \cdot h(T-1) + 1 - \alpha \leq c \cdot h(T) + 1,$$

as $h$ is nondecreasing, which is the desired result at $T$. In the second case, we just use $\text{err}_T \leq 1$, so

$$E_T \leq E_{T-1} + 1 - \alpha \leq c \cdot h(T-1) + 1 - \alpha \leq c \cdot h(T) + 1.$$

This again gives the desired result at $T$, and completes the proof. □

*Proof of Theorem 1.* We can transform (5) by setting $q'_{t+1} = q_{t+1} - \hat{q}_{t+1}$, and this becomes an update of the form (8) with respect to $q'_{t+1}$. Further, the score sequence in this new parameterization is $s'_t = s_t - \hat{q}_t$, which is in $[-b, b]$ because both $s_t$ and $\hat{q}_t$ are in $[-b/2, b/2]$. Applying Proposition 2 gives the result. □

*Proof of Proposition 3.* The proof is similar to that of Proposition 2—as in that proof, we only prove one side of the absolute inequality (11), and use induction. Abbreviate $E_T = \sum_{t=1}^{T} (L_i(\mathcal{C}_i, y_i) - \alpha))$. The base case holds trivially. For the inductive step, either $c \cdot h(T-1) < E_{T-1} \leq c \cdot h(T-1) + 1$ or $E_{T-1} \leq c \cdot h(T-1)$. In the first case, we have saturated, so $L_t(\mathcal{C}_t, y_t) = 0$, and

$$E_T = E_{T-1} - \alpha \leq c \cdot h(T-1) + 1 - \alpha \leq c \cdot h(T) + 1,$$

as $h$ is nondecreasing, which is the desired result at $T$. In the second case, we just use the boundedness of the loss $L_t(\mathcal{C}_t, y_t) \leq 1$, so

$$E_T \leq E_{T-1} + 1 - \alpha \leq c \cdot h(T-1) + 1 - \alpha \leq c \cdot h(T) + 1.$$

This again gives the desired result at $T$, and completes the proof. □

## B Comparisons to existing methods

This section contains descriptions of the relationship of our method to pre-existing methods for conformal prediction under distribution shift.

### B.1 ACI as a special case

Though it may not be immediately obvious, adaptive conformal inference (ACI) is actually a special case of the quantile tracker. ACI follows the iteration:

$$\alpha_{t+1} = \alpha_t - \eta(\text{err}_t - \alpha),$$

which is equivalent to

$$1 - \alpha_{t+1} = 1 - \alpha_t + \eta(\mathrm{err}_t - \alpha)$$
$$= 1 - \alpha_t + \eta \nabla \rho_{1-\alpha}(\beta_t - (1 - \alpha_t)),$$

for $\beta_t = \inf\{\beta : s_t \leq \mathrm{Quantile}_\beta(\{s_1, \ldots, s_{t-1}\})$. This shows that ACI is a particular instance of the quantile tracker that uses a secondary score $s'_t = \beta_t$ and quantile $q'_t = 1 - \alpha_t$. Thus, because quantile tracking (7) is the same as a linear coverage integrator (13), so is ACI.

We can see here that ACI transforms unbounded score sequences into bounded ones, which then implies long-run coverage for any score sequence. This may, however, come at a cost: ACI can sometimes output infinite or null prediction sets (when $\alpha_t$ drifts below 0 or above 1, respectively). Direct quantile tracking on the scale of the original score sequence does not have this behavior.

### B.2 Weighted conformal prediction

A line of work by [33] and [4] deals with variants of weighted conformal prediction. Specifically, the latter variant allows the incorporation of sample weights for each data point. When the data come from a probabilistic model, the sample weights allow for probabilistic bounds on the coverage of a new test point.

These methods, and others (e.g., [9]) in the probabilistic setting, differ greatly from the approach herein. When there is no probabilistic model, for example, all such approaches give vacuous coverage guarantees. The reason is that coverage on a *new test point* is impossible without a probabilistic model. However, in our setting we target long-run coverage, a completely different (and achievable) goal. Thus, neither the methods nor the proof techniques and theorem statements share any technical similarity to speak of.

For good measure, we include below in Appendix G evaluations against a version of [4] with a trailing window the size of the burn-in period. This approach generally performs worse than both our approach and ACI, and does not have guarantees in our setting, so we do not include it in main text.

### B.3 Sequential predictive conformal inference (SPCI)

A recently proposed line of work exemplified by Ensemble Batch Prediction Intervals (EnbPI) [38] and its improved version, SPCI [39], seeks to provide coverage guarantees for black-box time-series predictors. However, the methods therein are *not* distribution-free, and SPCI rely on the following assumptions for validity:

1. The use of the quantile random forest (QRF) algorithm.
2. Model assumptions, such as consistency.
3. Distributional assumptions, such as smoothness.

Thus, the validity of SPCI is significantly weaker than the methods herein, perhaps along the lines of a standard nonparametric analysis. In contrast, our method has finite-sample validity averaged over time for any input sequence. However, it is possible to use the QRF/SPCI algorithm as the scorecaster, which would endow it with a distribution-free guarantee. We do not do this because the QRF is not a particularly effective time series forecasting method, and was outperformed by quantile LASSO (which we use for the COVID experiments) and the Theta model. The performance of the QRF tends to be "blocky" and high-variance; for an example, see plots on the SPCI GitHub page.

However, we note that the EnbPI and SPCI algorithms can be seen as specific choices of scorecasters, and one consequence of our paper is that these algorithms can be endowed distribution-free guarantees when used as scorecasters in (5).

## C   Heuristics for setting constants

Consider the tan integrator $r_t(x) = K_I \tan(x \log(t)/(tC_{\mathrm{sat}}))$, where we set $\tan(x) = \mathrm{sign}(x) \cdot \infty$ whenever $x \notin [-\pi/2, \pi/2]$, and $C_{\mathrm{sat}}, K_I > 0$ are constants. The constant $C_{\mathrm{sat}}$ is primarily in charge of guaranteeing that by time $T$, we want to have an absolute guarantee of at least $1 - \alpha - \delta$ coverage.

Then we can set

$$C_{\text{sat}} = \frac{2}{\pi}\big(\lceil \log(T)\delta \rceil - 1/\log(T)\big)$$

to ensure the tan function has an asymptote at the correct point. The purpose of the constant $K_{\text{I}}$ is to place the integrator on the same scale as the scores. So if $B'$ is a hypothesized bound on the magnitude of the scores, then one can set $K_{\text{I}} = B'$. In practice, these heuristics can be taken as a starting place, and then the numbers can be fine-tuned during a burn-in period by hand or algorithmically. As alluded to previously, we believe there is room for work in the style of [17, 7] to rigorously tune these parameters online, but it is not the focus of our paper.

## D    Coverage integration as adaptive learning rate

As already mentioned in the introduction, in all our experiments we use a nonlinear saturation function $r_t(x) = K_{\text{I}}\tan(x\log(t)/(tC_{\text{sat}}))$, where we set $\tan(x) = \text{sign}(x) \cdot \infty$ for $x \notin [-\pi/2, \pi/2]$, and $C_{\text{sat}}, K_{\text{I}} > 0$ are constants that we choose heuristically (described in Appendix C). In a sense, this tan integrator is akin to a quantile tracker whose learning rate adapts to the current coverage gap. To see this, we can use a first-order Taylor approximation, which shows (ignoring constants):

$$q_{t+1} = \tan\left(\frac{\log(t)}{t}\sum_{i=1}^{t}(\text{err}_i - \alpha)\right) \approx q_t + \underbrace{\frac{\log(t)}{t}\sec^2\left(\frac{\log(t-1)}{t-1}\sum_{i=1}^{t-1}(\text{err}_i - \alpha)\right)}_{\text{effective learning rate}} \nabla\rho_{1-\alpha}(s_t - q_t).$$

Above, $\sec(x) = 1/\cos(x)$ is the secant function, which has a U-shape; thus we can see from the above that the effective learning rate is higher for larger errors. Similar analyses for different integrators will give different adaptive learning rates; see below for another example with a decaying learning rate.

Consider $r_t(x) = \eta x/\sqrt{t}$. (This will give long-run coverage only for bounded scores, because condition (4) is only met for finite and not infinite $b$.) Then (8) becomes $q_{t+1} = \frac{\eta}{\sqrt{t}}\sum_{i=1}^{t}(\text{err}_i - \alpha)$, which can be rewritten as

$$q_{t+1} = \frac{\sqrt{t-1}}{\sqrt{t}}\frac{\eta}{\sqrt{t-1}}\sum_{i=1}^{t-1}(\text{err}_i - \alpha) + \frac{\eta}{\sqrt{t}}(\text{err}_i - \alpha) \approx q_t + \frac{\eta}{\sqrt{t}}(\text{err}_t - \alpha).$$

This is approximately the quantile tracker (7) with a decaying learning rate, on the order of $1/\sqrt{t}$.

## E    Comparison to clipped ACI

Figures 7 and 8 compare the quantile tracker to a clipped version of ACI which disallows infinite-sized sets by clipping the sets to the largest score seen so far.

## F    More details on COVID-19 forecasting

In this experiment, the scorecaster receives as input the three most recent scores (i.e., quantile errors) of the ensemble forecaster, as well as the three most recent case and death counts, from *all 50 states*. The scorecaster is an $\ell_1$-penalized quantile regression as implemented by `sklearn.linear_model.QuantileRegressor`. We fixed tuning parameter for the $\ell_1$ penalty at 10; in our experience, the performance of the scorecaster was fairly robust to this choice. Automatic selection (e.g., using cross-validation) could be the topic of future study. Figures 9 and 10 shows the analogous experiments but for forecasting in New York and Texas.

## G    Further experiments

We give a more comprehensive view of our results, examing all data sets, and a range of tuning parameters for each method. We restrict our attention to AR as the base forecaster; for the rest of the base forecasters, we refer to the GitHub repository: `https://github.com/aangelopoulos/conformal-time-series`.

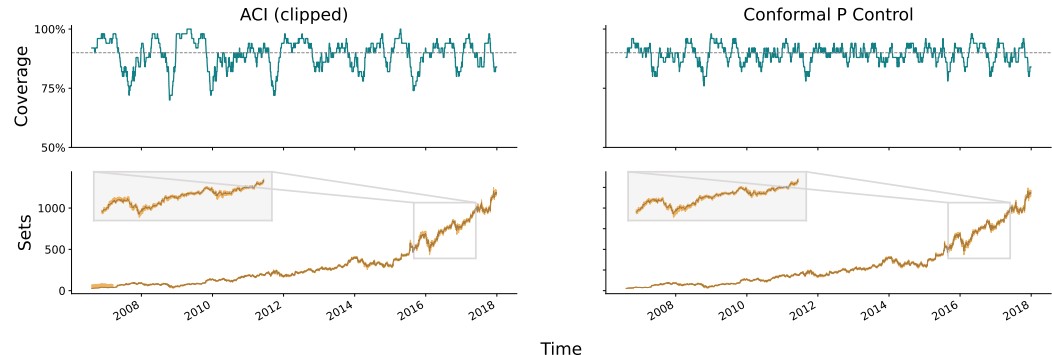

| | AR | | Prophet | | Theta | | Transformer | |
|---|---|---|---|---|---|---|---|---|
| | clip ACI | P Ctrl | clip ACI | P Ctrl | clip ACI | P Ctrl | clip ACI | P Ctrl |
| Marginal coverage | 0.898 | 0.898 | 0.884 | 0.897 | 0.898 | 0.896 | 0.884 | 0.897 |
| Longest err sequence | 2 | 2 | 8 | 3 | 2 | 2 | 7 | 3 |
| Average set size | 44.9 | 28.4 | 52.6 | 42.6 | 43.3 | 27.8 | 60.5 | 54.2 |
| Median set size | 41.8 | 18.6 | 38.8 | 31.9 | 27.3 | 18.7 | 36.6 | 35.5 |
| 75% quantile set size | 58.9 | 37.4 | 66.9 | 56.5 | 59.5 | 37.8 | 85.5 | 69.7 |
| 90% quantile set size | 93.9 | 66.3 | 137 | 93.4 | 94.7 | 63.4 | 148 | 123 |
| 95% quantile set size | 136 | 81.8 | 166 | 116 | 136 | 78.5 | 182 | 164 |

Figure 7: As in Figure 4, but with clipped ACI.

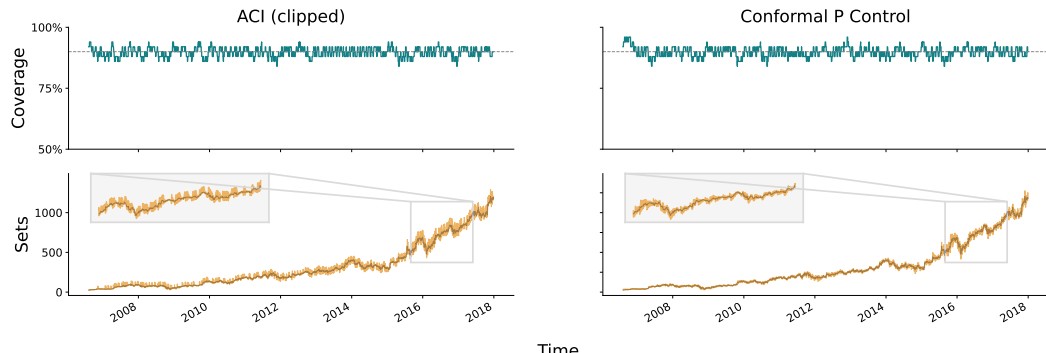

| | AR | | Prophet | | Theta | | Transformer | |
|---|---|---|---|---|---|---|---|---|
| | clip ACI | P Ctrl | clip ACI | P Ctrl | clip ACI | P Ctrl | clip ACI | P Ctrl |
| Marginal coverage | 0.894 | 0.894 | 0.896 | 0.888 | 0.895 | 0.894 | 0.89 | 0.887 |
| Longest err sequence | 6 | 3 | 13 | 6 | 5 | 5 | 21 | 9 |
| Average set size | 19.5 | 17.6 | 69.5 | 51.7 | 17.9 | 17.8 | 115 | 70.4 |
| Median set size | 13.4 | 13.3 | 48.7 | 37.3 | 12.8 | 13.1 | 61.7 | 44.3 |
| 75% quantile set size | 27.9 | 22.3 | 91.1 | 72.1 | 20.7 | 22.6 | 165 | 98.4 |
| 90% quantile set size | 44 | 37.7 | 168 | 114 | 38.5 | 38.4 | 248 | 153 |
| 95% quantile set size | 48.7 | 46.2 | 195 | 140 | 47.2 | 46.9 | 304 | 196 |

Figure 8: As in Figure 5, but with clipped ACI.

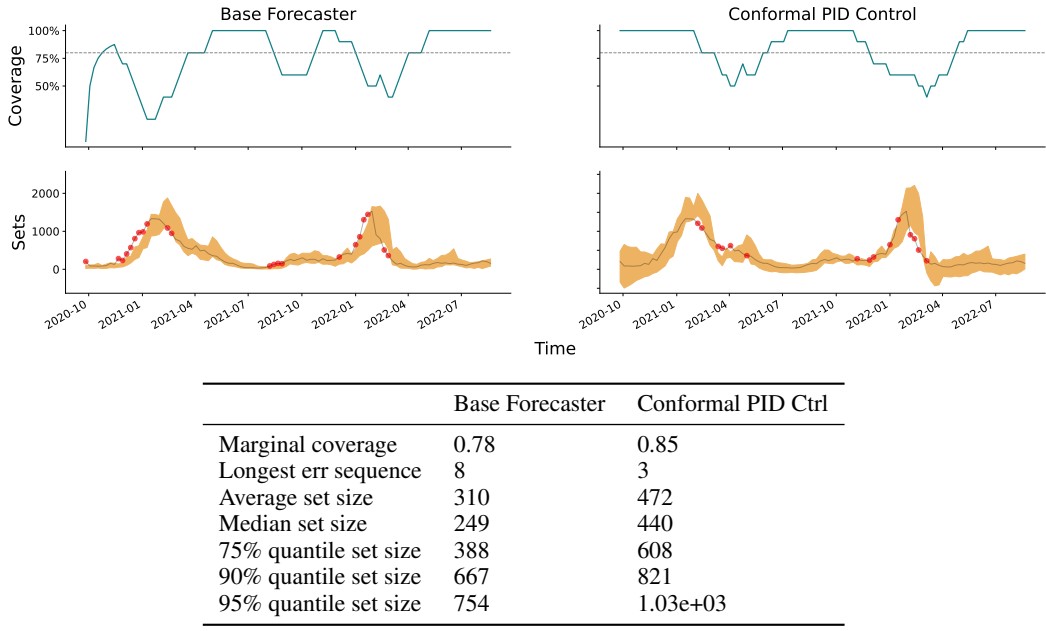

| | Base Forecaster | Conformal PID Ctrl |
|---|---|---|
| Marginal coverage | 0.78 | 0.85 |
| Longest err sequence | 8 | 3 |
| Average set size | 310 | 472 |
| Median set size | 249 | 440 |
| 75% quantile set size | 388 | 608 |
| 90% quantile set size | 667 | 821 |
| 95% quantile set size | 754 | 1.03e+03 |

Figure 9: Results for 4-week ahead COVID-19 death forecasting in New York.

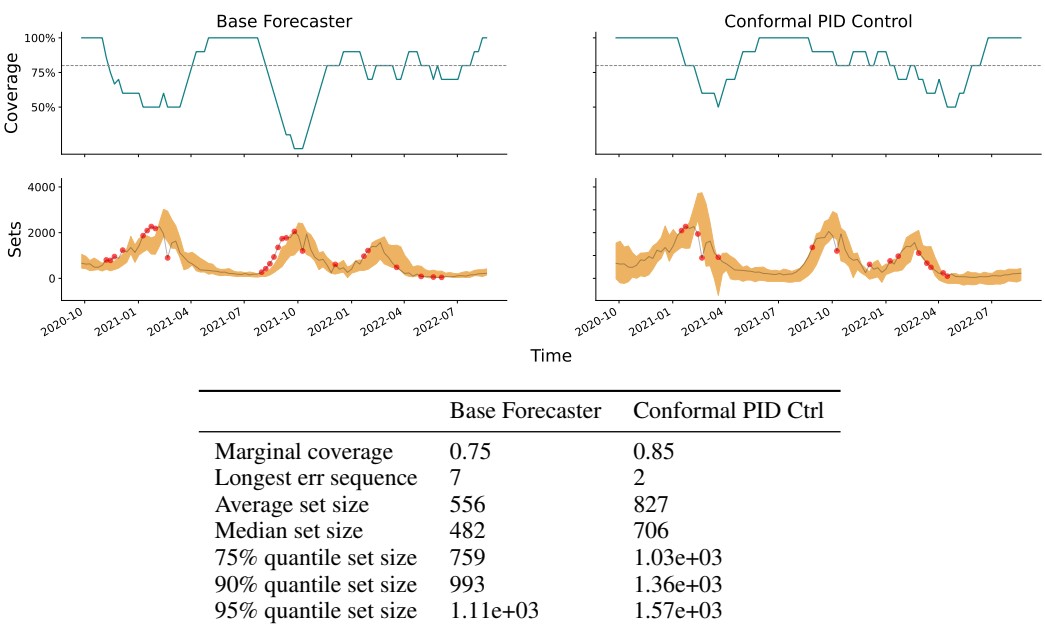

| | Base Forecaster | Conformal PID Ctrl |
|---|---|---|
| Marginal coverage | 0.75 | 0.85 |
| Longest err sequence | 7 | 2 |
| Average set size | 556 | 827 |
| Median set size | 482 | 706 |
| 75% quantile set size | 759 | 1.03e+03 |
| 90% quantile set size | 993 | 1.36e+03 |
| 95% quantile set size | 1.11e+03 | 1.57e+03 |

Figure 10: Results for 4-week ahead COVID-19 death forecasting in Texas.

|                      | Base Forecaster | Conformal PID Ctrl |
| -------------------- | --------------- | ------------------ |
| Marginal coverage    | 0.82            | 0.86               |
| Longest err sequence | 6               | 2                  |
| Average set size     | 625             | 858                |
| Median set size      | 512             | 688                |
| 75% quantile set size | 754            | 1.01e+03           |
| 90% quantile set size | 1.12e+03       | 1.47e+03           |
| 95% quantile set size | 1.45e+03       | 1.8e+03            |

Table 1: Summary statistics for COVID-19 death forecasting in California, as in Figure 2.

|                      | AR | | | Transformer | |
|                      | ACI | Conformal PID Control | | ACI | Conformal PID Ctrl |
| -------------------- | ----- | ------------------- | - | ----- | ------------------ |
| Marginal coverage    | 0.899 | 0.9                 |   | 0.899 | 0.901              |
| Longest err sequence | 3     | 2                   |   | 3     | 2                  |
| Average set size     | $\infty$ | 0.177            |   | $\infty$ | 0.174           |
| Median set size      | 0.406 | 0.178               |   | 0.426 | 0.175              |
| 75% quantile set size | 0.484 | 0.21               |   | 0.574 | 0.206             |
| 90% quantile set size | 0.672 | 0.236              |   | $\infty$ | 0.233          |
| 95% quantile set size | $\infty$ | 0.252           |   | $\infty$ | 0.249          |

Table 2: Summary statistics for electricity forecasting, as in Figure 3. Results for the Prophet and Theta models are not available because `darts` does not support intermittent retraining for these algorithms.

For each experiment, we describe the data set in a new subsection, and two plots are included: one for the coverage, and one for the prediction sets. Each column in the plots represents a different method, and each row is a different learning rate. For the quantile tracker, the learning rate is to be interpreted as the multiplier in front of $\hat{B}_t$. Each method is given a different color, which stays consistent throughout the plots. We use a tan integrator and a Theta scorecaster throughout, just as in the main text experiments. There is an additional method, 'Trail', corresponding to the proposal of Barber et al. [4] with a trailing window.

### G.1 Amazon/Google

These data sets are part of a multivariate time series consisting of thirty blue-chip stock prices, including those of Amazon (AMZN) and Google (GOOGL), from January 1, 2006 to December 31, 2014. We attempt to forecast the daily opening price of each of Amazon and Google stock, on a log scale. Available to the scorecaster are the previous open prices of *all 30 stocks*.

### G.2 Microsoft

This data set is a univariate time series consisting of a single stock open price, that of Microsoft (MSFT), from April 1, 2015 to May 31, 2021.

### G.3 Daily temperature in Delhi

This data set contains the daily temperature (averaged over 8 measurements in 3 hour periods), humidity, wind speed, and atmospheric temperature in the city of Delhi from January 1, 2003 to April 24, 2017, scraped using the Weather Underground API.

### G.4 Electricity demand forecasting

This data set measures electricity demand in New South Wales collected at half-hour increments from May 7th, 1996 to December 5th, 1998 (we zoom in on the first 2000 time points). There are also

several other variables collected, such as the demand and price in Victoria, the amount of energy transfer between New South Wales and Victoria, and so on. These are given as covariates to the scorecaster. The demand value is normalized by default to lie in $[0, 1]$.

### G.5 Synthetic data sets

We perform some experiments on two synthetic score sequences which include change points and other behaviors difficult to produce using real data. In this setting, there is no ground truth $y_t$ sequence, so we do not plot the sets. Instead, we plot the scores themselves in one column, and the quantiles $q_t$ produced by each algorithm in a different column (when $q_t \geq s_t$, we cover). The general goal is for $q_t$ to track the $1 - \alpha$ quantile of $s_t$, and if it is too far off, that corresponds to the "set being too large or too small" in a situation where we would be constructing sets out of these scores.

We consider an i.i.d. sequence of scores, a noisy increasing sequence of scores, and a mix of change points and trends. Our codebase describes the score generation procedure in more detail.

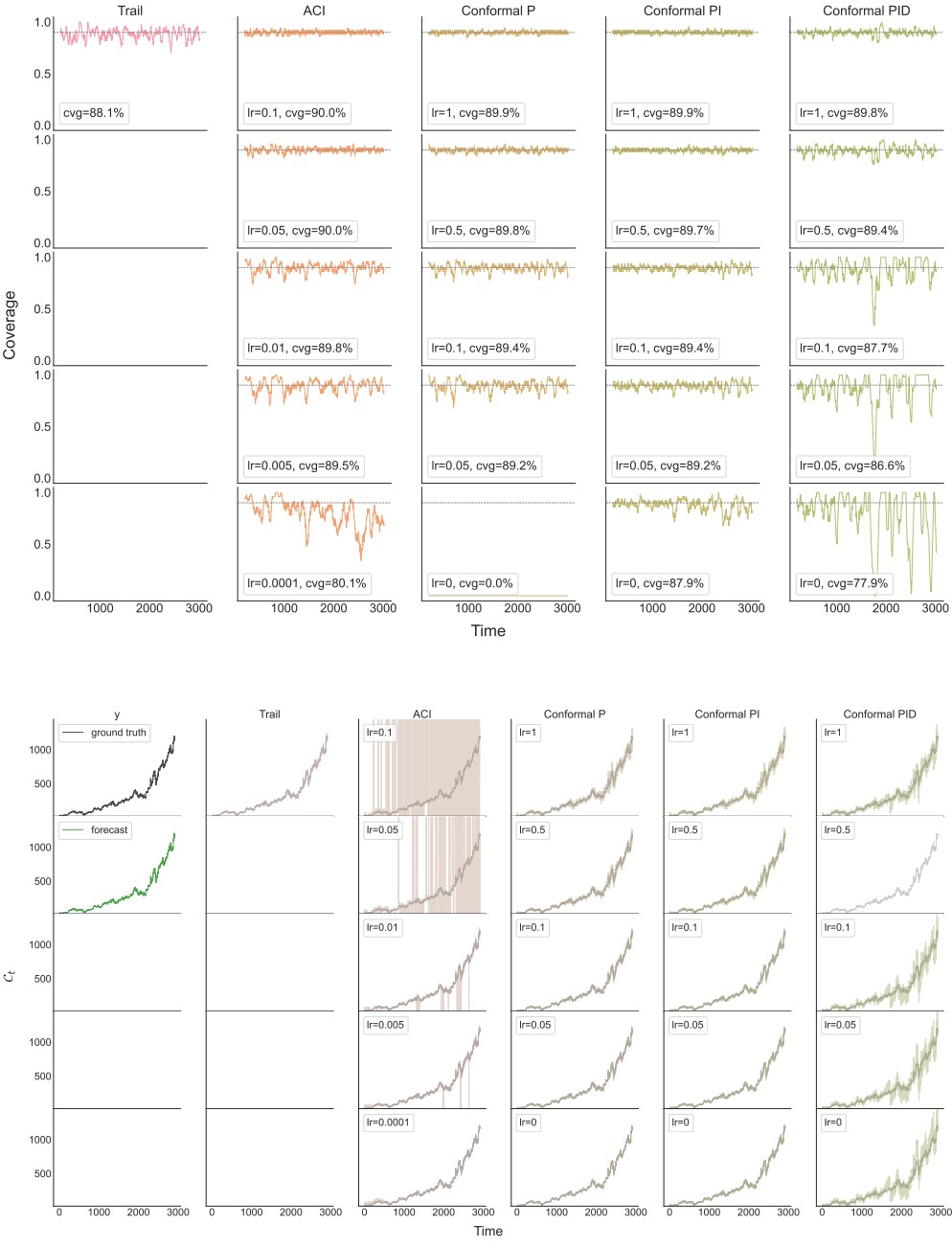

Figure 11: Results for the Amazon data set.

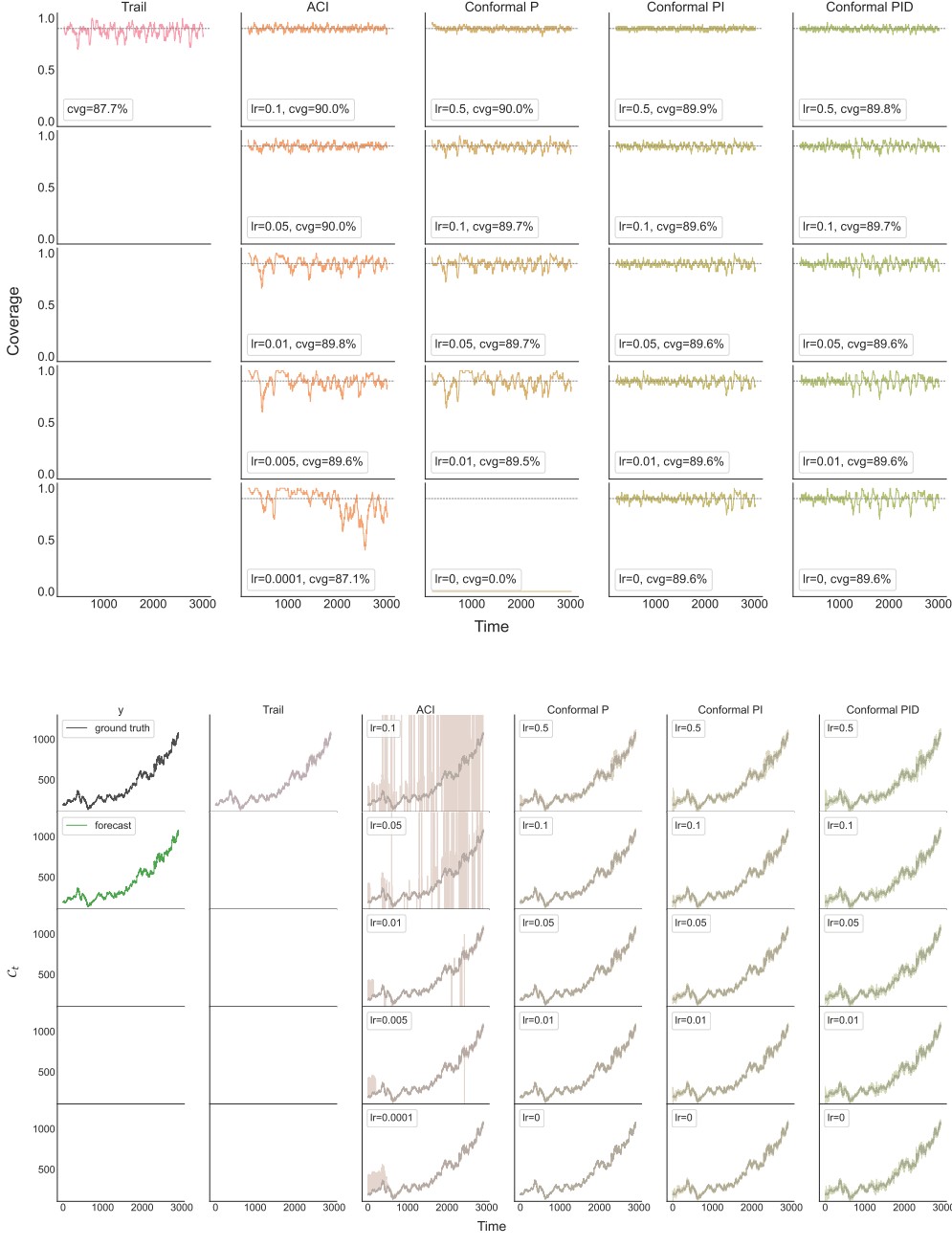

Figure 12: Results for the Google data set.

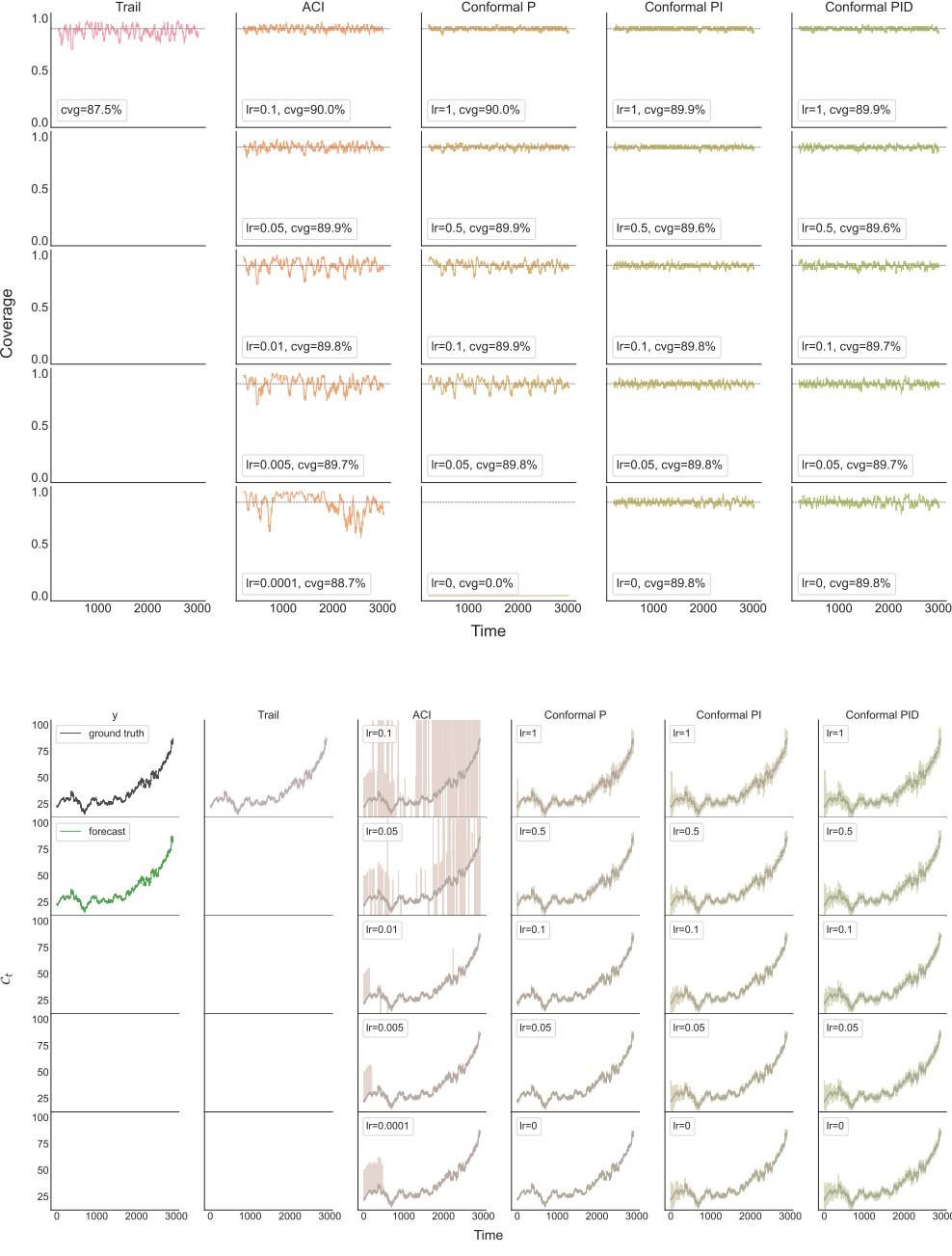

Figure 13: Results for the Microsoft data set.

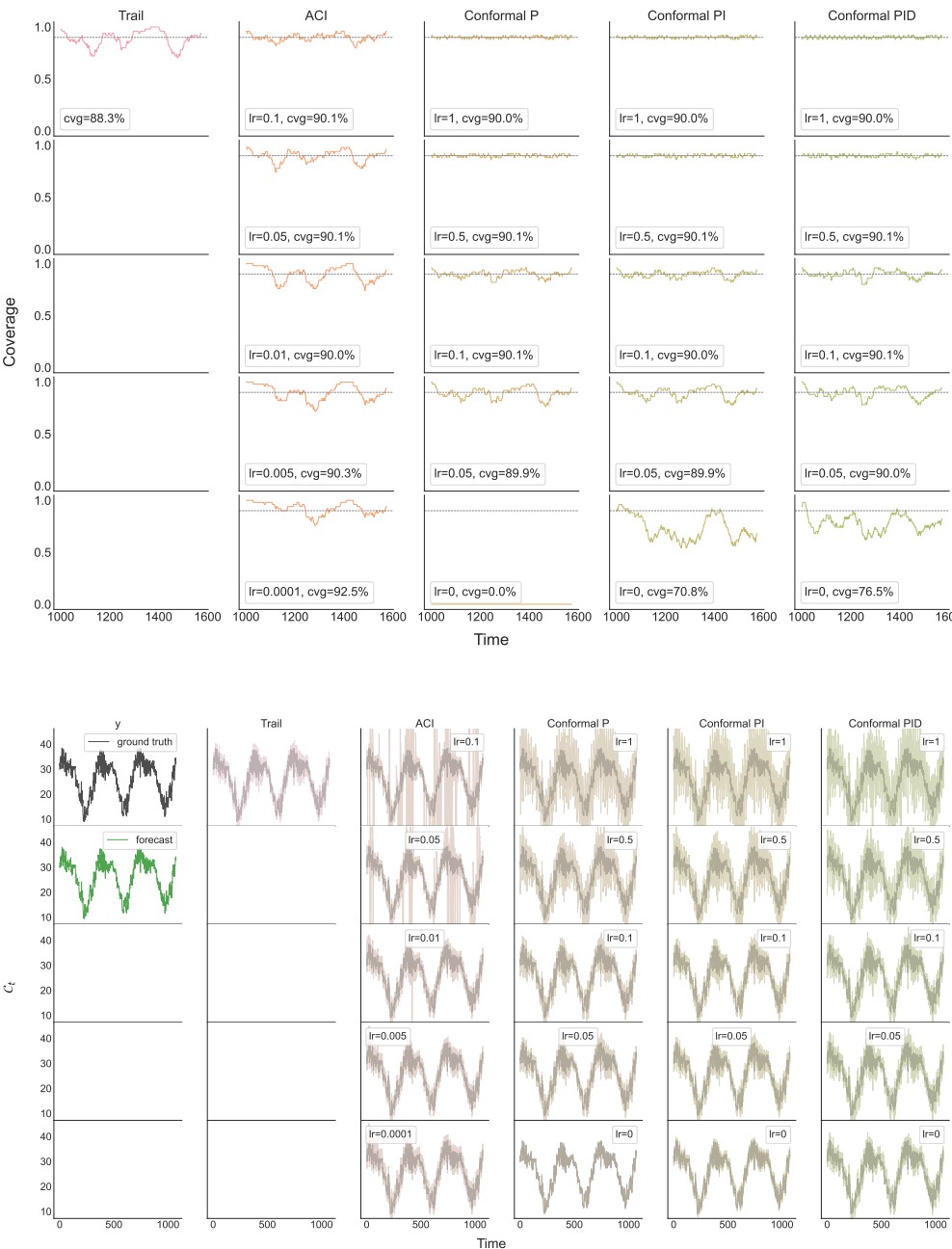

Figure 14: Results for the Delhi temperature data set.

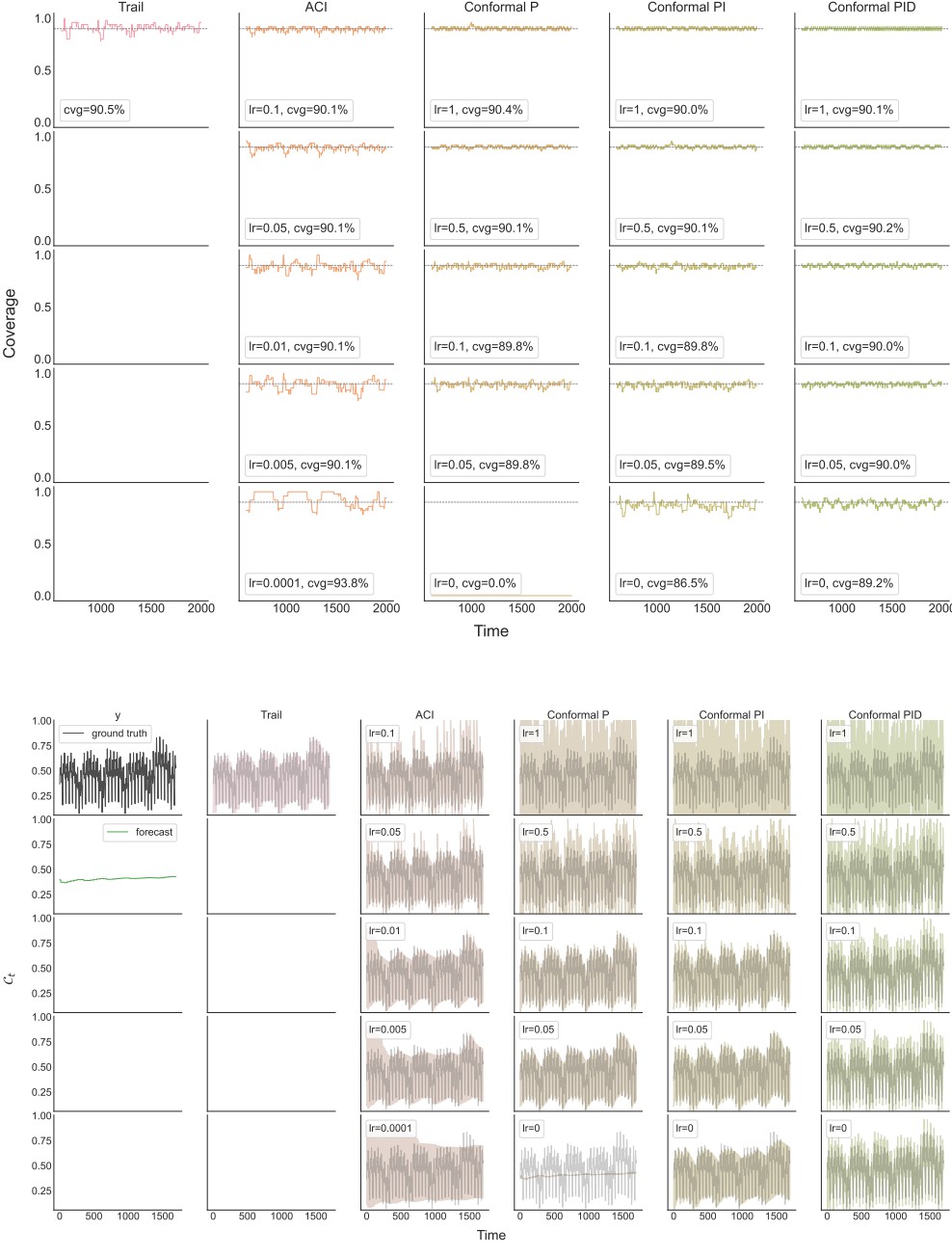

Figure 15: Results for the electricity demand forecasting data set.

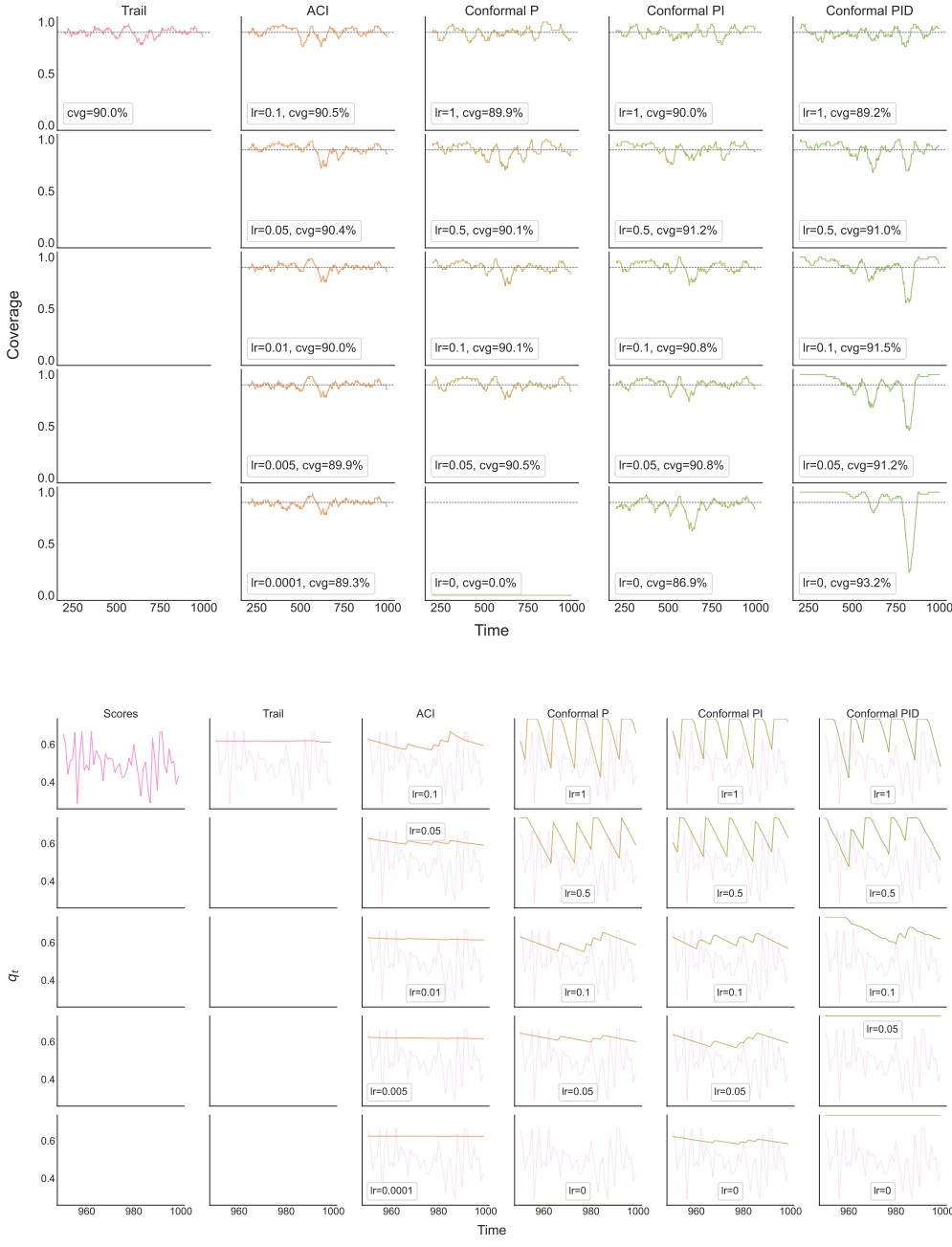

Figure 16: Results for an i.i.d. score sequence.

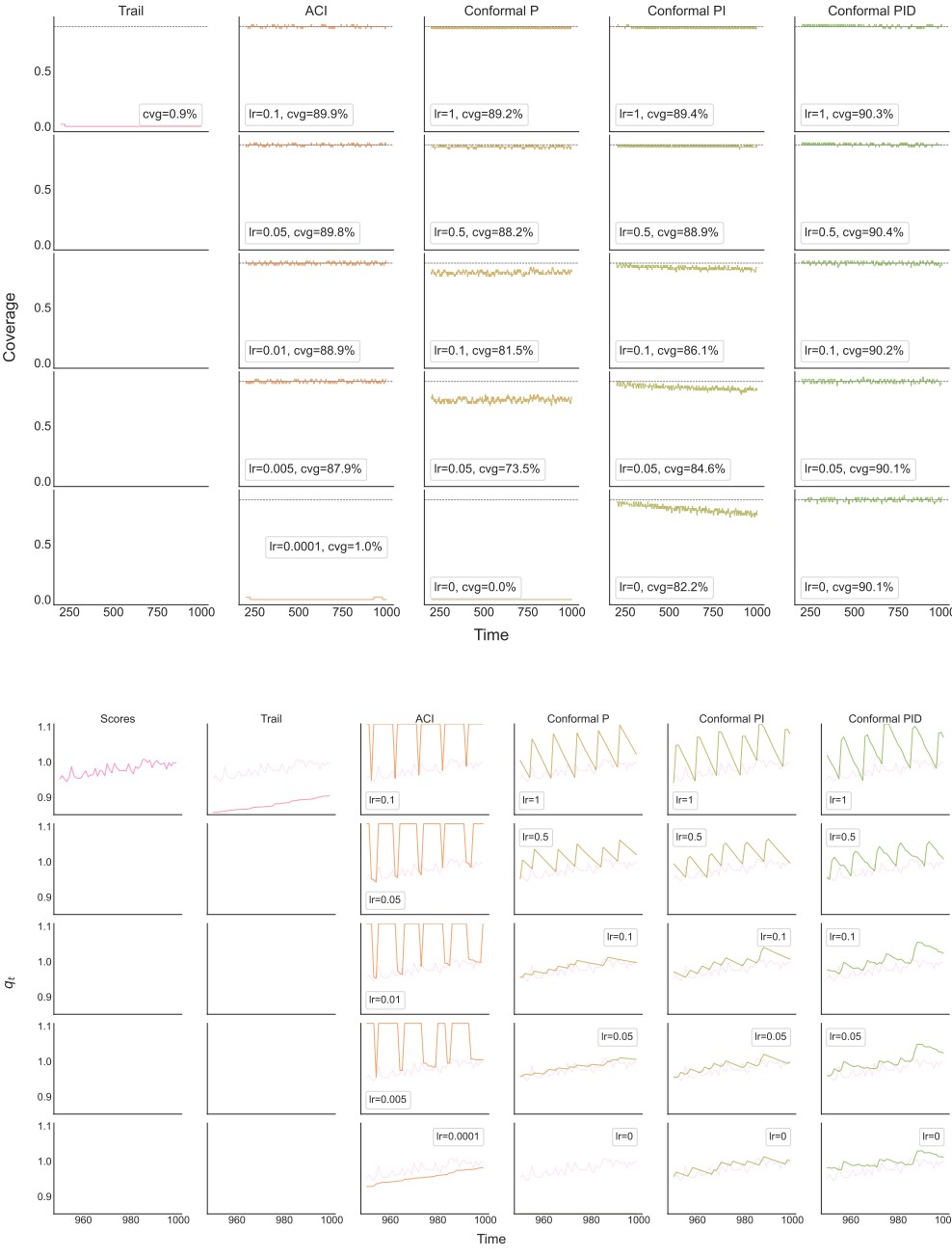

Figure 17: Results for an increasing score sequence.

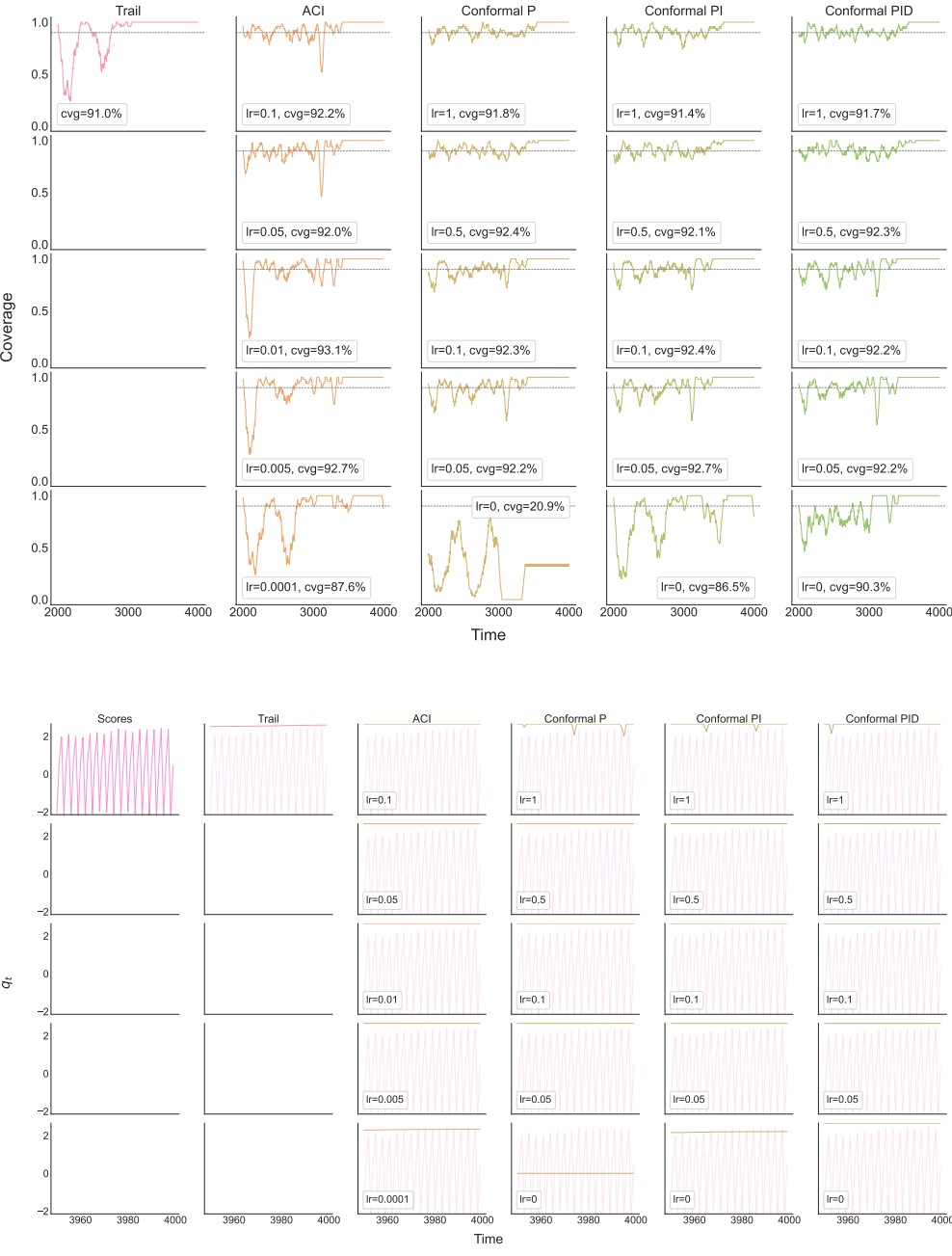

Figure 18: Results for a score sequence that is a mix of change points and trends.

