# OpenReview forum: "Conformal PID Control for Time Series Prediction"
_NeurIPS.cc/2023/Conference — NeurIPS 2023 poster_

### Official Review · Reviewer_H9oS · 2023-07-04

**Soundness:** 4 excellent
**Presentation:** 4 excellent
**Contribution:** 3 good
**Rating:** 6
**Confidence:** 4

**Summary:**

The paper proposes an adaptation to existing adaptive conformal prediction in two novel ways: (i) by tracking the quantile via online regression over the running sum of the errors; and (ii) by incorporating a second model to anticipate the quantile in the next instant ("scorecasting"). The authors include theoretical guarantees for the resulting method, as well as comparisons against a benchmark in the field (ACI, which is shown to be a a special case of the proposed algorithm) over many different datasets.

**Strengths:**

- Clear: the paper is well-written and very clear regarding its contributions.
- Limitations: the discussions on the method's limitations are very helpful.
- Theoretical results: the mathematical proofs are easy to follow and the notation is clear.
- Empirical results: the empirical studies are extensive and cover several different datasets. The experiments in the supplementary material are also very thorough.
- Code: the code is well-organized and readable (but please include the `environments.yml` file, all the necessary datasets, and instructions on how to reproduce figures in the paper! See below).

**Weaknesses:**

- Theoretical guarantees: the guarantees of the proposed method are only asymptotic. Unfortunately, no finite-sample results are included. None of the theoretical developments in the paper showcase advantages that the proposed method has over leading alternatives.
- Baselines: there are many recent proposals in the field (even in your related works section), but the only baseline is ACI. SPCI (Xu and Xie, 2022) also obtained asymptotic longitudinal coverage of time series without creating infinitely-wide intervals; can you underline in what ways your proposal is better?
- Experimental results: Can you include more details on how the hyperparameters were chosen for the examples in the main paper (for example, $r_t$)? Having too many hyper parameters can be overwhelming to the end-user (I appreciate the discussion in appendices C and D!). It would be great to include at least one example of adaptive learning in the proposed algorithm.
- Scalability: all the time series considered, both in the main text and in the supplementary material (SM), are small, with the largest ones comprising around 3000 data points. In scenarios where new data arrives every second, for example, it is not clear if the method would work as-is or if any modifications would be needed. A discussion about this and an experiment with a very long time series would be great.
- References: the authors point out the field of online calibration, and mention connections to game theory and online learning, but never explicitly discuss the connection to their work. It would be nice to better understand what tools from the field you are using or building upon.
- Relationship to control theory: seems slightly out-of-place. Either explain the connection better (perhaps in the SM) or develop it in another paper? As it is, it's hard for the CP audience understand what the PID is, or why this connection is relevant.
- Reproducibility: adding the proposed method's pseudocode would help a lot, either to the main text or to the SM if space is lacking. Trying to reproduce the results following the README.md, I was instructed to install the dependencies listed on `environment.yml`, but no such file was provided and I had to track them down one by one. There also seems to be a dataset missing (`datasets/preds_cases.csv`) and not all plots from the paper were generated.
- Minor writing suggestion: on one hand, the relation to control theory seems currently very incipient and more of a curiosity, but it is presented prominently in a section of its own. On the other hand, the interesting Proposition 3 is cramped in the final section. You might consider adding an independent section for extensions and incorporate the control theory analogy into the final discussion section.
- Other minor suggestions: I don't see the point of including the zoom in the figures; it oftentimes hides parts of the figure and the underlying patterns are clear from the whole graph. Besides the many plots, adding a table with a summary of the results among all datasets could help a reader to draw conclusions faster. The meaning of sublinear function is given in line 164, but the term is used before in lines 55 and 67, so we suggest bringing the definition up. Display (5) has a floating \cdot in the last inequality and inconsistent usage (c \cdot h(t) vs -ch(t)). Figure 1 caption says prediction sets are in purple, but it is actually in orange.
- Typos: line 115: $(1-\alpha)$ should be $\tau$; line 139: the em-dash is confusing.; line 151: "an particular" should be "a particular; proof of proposition 2: $C$ should be $c$; display under line 192: the second $\text{err}_i$ should be $\text{err}_t$; line 201: "can made" should be "can be made"; line 295: "saturation function as in 2" should be "saturation function that satisfies (5)" or "saturation function as in Proposition 2"line 301: missing \mathrm for err_i.; the bibliography is not printed in the SM and the references do not match the main text, so it ca be confusing.

**Questions:**

- Why is asymptotic coverage the best one can hope for in the adversarial setting? Is this a formal result? It would be helpful to develop this idea a bit more in the text.
- What are advantages of using the scorecaster in the conformal prediction setup vs adapting the model instead?
- Any guidance on how should one pick $r_t$ and the learning rate? How was this decided for the figures in the main paper? The paper seems to imply trial and error. The impact on figures in the appendix is extremely large!
- When do you expect ACI to be better than Quantile+Integrator+Scorecaster?
- What are advantages and disadvantages of your proposal vs SPCI, given both achieve coverage of time series without creating infinitely-wide intervals? How does the empirical performance compares? In what ways is your proposal better?
- Can you explain what you mean by "Our algorithms may not be universally useful on time series different from those we used for evaluation" (line 283)? When should it be useful?
- There is a method called "Trail" in the appendix plots, but no explanation. What is it?

**Limitations:**

Yes.

---

> ### Author Rebuttal · Authors · 2023-08-10
>
> We thank the reviewer for the accurate summary, and the engagement (esp. on reproducibility!)
>
> See the main response (in a separate comment) for a summary of the extensive improvements we have made to the paper. Below we reference the specific points related to your critique. (We have shortened/paraphrased some questions for space.)
>
> > Code and reproducibility concerns: please include the environments.yml file, datasets, and instructions.
>
> So sorry for the omission! We have made the code much more usable (added requirements.txt file, instructions, all datasets), towards public release.
>
> > Guarantees: guarantees are only asymptotic. No finite-sample results are included.
>
> In the new version of the paper, we have added finite-sample bounds for all algorithms; see the main response comment (2). In addition, existing work such as Xu and Xie does not have a distribution-free guarantee. We elaborate in the main response comment (5), and will add a new section to the camera-ready.
>
> The Xu and Xie result requires the use of a specific model (QRF), and distributional smoothness assumptions leading to, e.g., model consistency. It is also only asymptotic.
>
> Our results have finite-sample validity for adversarial sequences.
>
> Please see the main responses (2) and (5) for further details.
>
> > Experimental results: adaptive learning rate?
>
> Yes, definitely! In the new paper, we have provided a default choice for all parameters; see the main response (3).
>
> > Scalability
>
> Great question. The reviewer makes a great point that sometimes samples are collected every second or even faster.
>
> For extremely long time series that require extremely fast predictions (i.e. with a sub-millisecond latency), such as those in high-speed-trading, we may need some modifications of our methods. Quantile tracking would work as-is, with no changes. The windowed version of integral control might be more appropriate for the extremely long sequence. Finally, efficient scorecasting algorithms would be needed to ensure fast prediction.
>
> We will add a paragraph in the discussion section mentioning this as an opportunity for future work in which non-obvious limitations of our approach might appear. The challenge of building a system of this scale and speed is likely to be an interesting and independent project.
>
> Thank you for the wonderful point. It is a really nice suggestion.
>
> > References: It would be nice to better understand what tools from online calibration you are using or building upon.
>
> The online learning community works on problems in the adversarial sequence model, but they do not provide guarantees of the sort studied in our work. We do not build upon the technical tools from that area — and we have tried to survey that literature for close connections, to little avail.
>
> We will mention this in the new “Comparison to Existing Methods” section. The quantile tracker is equivalent to online gradient descent, an algorithm well-studied in online optimization. However, regret guarantees cannot be used to prove guarantees of the same form as in our work. The idea of what we call “saturation” does not seem to appear in that literature (perhaps because it isn’t natural for systems that are not prediction sets), and it is the core idea of our proofs.
>
> > Relationship to control theory: seems slightly out-of-place.
>
> Totally agreed. In the new rewrite, we have made this connection far more precise and formal, starting in the introduction.
>
> > Add pseudocode
>
> The idea of pseudocode is great! We will add it to the supplement, since we’re already running out of space, but we will reference it very clearly in the main text.
>
> > Add exensions section?
>
> We’ll create a dedicated extensions section; see the main response (4).
>
> On the note of control, we agree that having its own section was overkill. At the same time, we have made the connection more precise to the point that we feel it can be more of a motivational point, so we have added it to the introduction and removed the dedicated section.
>
> > Why is asymptotic coverage the best one can hope for in the adversarial setting?
>
> Great question. The original statement is actually not fully true (c.f. finite sample results). For a proof sketch of the impossibility of coverage at time $t$ (as opposed to the average version), consider the following.
>
> Conformal picks any quantile $q_{t+1}$.
>
> The adversary sees $q_{t+1}$, and picks $s_{t+1} = q_{t+1} + \epsilon$.
>
> Then conformal miscovers, unless $q_{t+1} = \infty$ with some probability $\alpha$.
>
> Therefore, conformal fails at any time $t$ without randomization. So we instead need to target an averaged form of coverage. This is not far from a formal argument. We would be happy to write it down at your request.
>
> > Any guidance on how should one pick and the learning rate?
>
> See the main response comment (3).
>
> > What are advantages of using the scorecaster in the conformal prediction setup vs adapting the model instead?
>
> If the base model can be adapted easily, it might be better to do that. But sometimes it is hard to adapt the base model (for engineering or political reasons), or systematic trends in the scores persist even after extensive engineering. E.g., the covid model was trained by the CDC, but still can be improved by our method!
>
> > When do you expect ACI to be better than Quantile+Integrator+Scorecaster?
>
> If very bad choices of parameters are chosen for the integrator, or the scorecaster is really badly trained and actually adds error, then ACI can definitely be better. (We have added a sentence to this effect.)
>
> > Comparison to SPCI?
>
> See the main response comment (5).
>
> > There is a method called "Trail" in the appendix plots, but no explanation. What is it?
>
> This is Barber et al. with a trailing window; see main respons  (5).
>
> > Besides the many plots, adding a table with a summary of the results among all datasets could help a reader to draw conclusions faster.
>
> See (1) of the response.
>
> > Minor points
>
> Thank you! Very helpful.

---

> > ### Comment · Reviewer_H9oS · 2023-08-16
> >
> > Thank you for your informative response and the valuable additions to the paper!

---

> > > ### Author Response · Authors · 2023-08-17
> > >
> > > Thank you for the feedback! We're glad to hear that our revisions have been well-received, and hope to have earned your support of our paper's acceptance. Please let us know if there is anything more we can do to further address your concerns or provide additional clarity.

---

### Official Review · Reviewer_bvK1 · 2023-07-08

**Soundness:** 3 good
**Presentation:** 3 good
**Contribution:** 3 good
**Rating:** 7
**Confidence:** 4

**Summary:**

The paper tackles the problem of conformal UQ under distribution shifts. This is a very realistic setting where the upstream prediction model cannot be frequently updated, and we still want to certify some level of safety. The paper cleverly fames the problem as a PID control problem, and provides a practical algorithm that they show is effective on synthetic and real datasets.

**Strengths:**

- The setting addressed is impactful and practical. A major hindrance of using CP in ML systems deployment is that the exchangeability assumption is often violated. This work provides a practical adaptation while still maintaining some guarantees of the method.

- The PID framing of the problem & solution is clever and insightful. (very Big Brain, if i may use gen-z lingo.) The two Gibbs & Candes paper from 2021 and 2022 for CP under distribution shifts was able to circumvent the exchangeability assumption and achieve some guarantees by transforming the distribution shift into a online optimization problem for a single parameter $\alpha^*$, which they prove is similar to a P controller.  This work takes the idea a step further, and introduces the integrater (I) and forecasting (D) algorithms, completing the picture (fig 2 <- such a nice figure).

- The method can utilize time-series modeling approaches to adapt to seasonality.

- The experiment section is well-presented. It walks you through part-by-part of the PID controller and shows the impact of each. The examples chosen are clear and convincing.



**Weaknesses:**

1. The paper did not define some concepts it used in its writing. For example:
- line 29, exchangeability
- line 25, sharpest
- line 83, burn-in period
- line 137 (not really a definition just... what are you trying to say here?) "the quantile tracker proceeds agnostically and performs the same updates in any case."

I think explaining them, even just in the appendix, will help with clarity.

2. The paper only compared to ACI & ablations of its own method. A common choice of algorithm for the scenario introduced by the authors would be Enbpi and SPCI from Xu & Xie (references 29 and 30 in the paper).

**Questions:**

- Can you elaborate on the reason why Enbpi and SPCI from Xu & Xie are not good baselines for comparing your model? Many have used previous timesteps as covariate X, similar to your scorecasting approach.

**Limitations:**

The authors provided a good address of limitations in their discussion section.

---

> ### Author Rebuttal · Authors · 2023-08-10
>
> > The paper tackles the problem of conformal UQ under distribution shifts. This is a very realistic setting where the upstream prediction model cannot be frequently updated, and we still want to certify some level of safety. The paper cleverly frames the problem as a PID control problem, and provides a practical algorithm that they show is effective on synthetic and real datasets.
>
> We thank the reviewer for their kind comments and helpful suggestions. We are interested in the reviewer’s thoughts on our expansion/revision of the paper as well.
>
> Please see the main response (in a separate comment) for a summary of the extensive improvements we have made to the paper. Below we reference the specific points related to your critique.
>
> > The setting addressed is impactful and practical. A major hindrance of using CP in ML systems deployment is that the exchangeability assumption is often violated. This work provides a practical adaptation while still maintaining some guarantees of the method.
> The PID framing of the problem & solution is clever and insightful. (very Big Brain, if i may use gen-z lingo.) The two Gibbs & Candes paper from 2021 and 2022 for CP under distribution shifts was able to circumvent the exchangeability assumption and achieve some guarantees by transforming the distribution shift into a online optimization problem for a single parameter, which they prove is similar to a P controller. This work takes the idea a step further, and introduces the integrator (I) and forecasting (D) algorithms, completing the picture (fig 2 <- such a nice figure).
>
> We thank the reviewer! We also enjoyed making Figure 2 :)
>
> > The paper did not define some concepts it used in its writing. For example... (lists examples)
>
> We appreciate the fixes. We can and will make edits to these places for clarity.
>
> > The paper only compared to ACI & ablations of its own method. A common choice of algorithm for the scenario introduced by the authors would be Enbpi and SPCI from Xu & Xie (references 29 and 30 in the paper). Can you elaborate on the reason why Enbpi and SPCI from Xu & Xie are not good baselines for comparing your model? Many have used previous timesteps as covariate X, similar to your scorecasting approach.
>
> Absolutely. We agree that the previous methods by Xu and Xie share a relationship to scorecasting. The main difference is that these methods do not have a distribution-free guarantee, while ours does. Indeed, the method of Xu and Xie can be used as a scorecaster — but the scorecasting framework works for a larger class of methods, including more advanced ones that might not have the guarantees from that paper. This is explained in more detail in _main response point 5._
>
> That said, we can include a new subsection, titled “Comparison with Existing Methods.” This goes through the related literature in greater detail, and explains the exact mathematical connections — including which methods we chose to compare to, and why.
>
> We hope this makes the relationship between our new results and related works clearer.

---

### Official Review · Reviewer_aXqf · 2023-07-08

**Soundness:** 2 fair
**Presentation:** 3 good
**Contribution:** 3 good
**Rating:** 6
**Confidence:** 4

**Summary:**

This paper examines the issue of how to parameterize conformal prediction in the time series setting. It argues that standard conformal inference methods would not provide valid inferences in the sequential setting, which lacks exchangeability. To resolve this, the paper suggests using online quantile tracking, integrating errors for stabilizing coverage, and utilizing scorecasting in the presence of systematic trends. The authors establish the corresponding theoretical properties, such as asymptotic coverage and conduct several experiments to justify their approach.

**Strengths:**

1. A new method for conformal prediction in the in the time series setting with a clear presentation of the details and background.
2. The authors show the proposed method has a major practical benefit, that unlike ACI (Gibbs and Candes, 2021), no infinite sets are produced.
3. The authors demonstrate the practical utility of their methods with an extensive series of experiments on a wide range of real-time series data sets.

**Weaknesses:**

1. Although this paper present a general theorem that their method achieves asymptotic coverage, there is insufficient discussion on the coverage guarantees such as the lower bounds and upper bounds on coverage.
2. The authors also explicitly avoid addressing the choice of automatic algorithms for the
two parameter $C_{sat}$ and $K_I$ in the nonlinear saturation function.
3. In the sequential setting (7), the authors didn’t show how to select the learning rate
$\eta$, what factors might affect the selection of the learning rate $\eta$, and what impact this
selection have on the performance of the method.

**Questions:**

1. How the specific selection of the function $r_t$ and $\hat{q}_t$ influence the
overall performance of the proposed method in conformal prediction for time series?
2. In the context of time series conformal prediction, how does this new method compare
with other popular methods in terms of robustness, and computational efficiency?
3. How does a non-constant threshold function $h$ help with tolerating a greater degree of
coverage error throughout the sequence?

**Limitations:**

1. Despite demonstrating strong results across various experiments, the authors acknowledge
potential limitations, including the challenge of proving asymptotic coverage, potential
inapplicability to different time series, and limitations in scorecasting.
2. The methods they develop are specific to cases where response $y_t$ is revealed at each time
point. However, there are many settings in which we receive the response in a delayed
fashion or in large batches.

---

> ### Author Rebuttal · Authors · 2023-08-10
>
> > This paper examines the issue of how to parameterize conformal prediction in the time series setting. It argues that standard conformal inference methods would not provide valid inferences in the sequential setting, which lacks exchangeability. To resolve this, the paper suggests using online quantile tracking, integrating errors for stabilizing coverage, and utilizing scorecasting in the presence of systematic trends. The authors establish the corresponding theoretical properties, such as asymptotic coverage and conduct several experiments to justify their approach.
>
> Thank you for the summary, and we appreciate your engagement.
>
> Please see the main response (in a separate comment) for a summary of the extensive improvements we have made to the paper. Below we reference the specific points related to your critique. We hope our responses answer the questions sufficiently to earn the reviewer’s support. Please let us know how we can improve if you still feel we are missing something.
>
> > Although this paper presents a general theorem that their method achieves asymptotic coverage, there is insufficient discussion on the coverage guarantees such as the lower bounds and upper bounds on coverage.
>
> In the new version of the paper, we have included finite-sample bounds that exactly quantify the lower and upper bounds on coverage; see the main response (2).
>
> > The authors also explicitly avoid addressing the choice of automatic algorithms for the two parameter and in the nonlinear saturation function.In the sequential setting (7), the authors didn’t show how to select the learning rate, what factors might affect the selection of the learning rate, and what impact this selection have on the performance of the method.
>
> We fully agree. In the new version, we have added default parameter selections for all parameters; see the main response (3).
>
> > How the specific selection of the function r and h influence the overall performance of the proposed method in conformal prediction for time series? How does a non-constant threshold function help with tolerating a greater degree of coverage error throughout the sequence?
>
> We can add a new paragraph in our methods section addressing this point. The function $h(t)$ (which is the main determinant of $r(t)$) parametrizes a tradeoff between coverage and smoothness. For any function $h(t)$ the coverage gap will be no more than $(h(t)+1)/T$. Here are some example choices:
> Taking $h(t) = c$ enforces that, for all $T$, the long-run coverage $1-\frac{1}{T}\sum_{t=1}^T \mathrm{err}_{t}$ is within $1-\alpha \pm (c+1)/T$.
> If the function $h(t)$ grows (super-)linearly, asymptotic coverage is not guaranteed, but it can still be achieved approximately. For example, taking $h(t) = cT$ enforces that the same average coverage as above is within $1-\alpha \pm (c + 1/T)$.
> However, if the user sets, say, $h(t) = \sqrt{T}$, then the long-run coverage is within $1-\alpha \pm \frac{\sqrt{T} + 1}{T} \approx 1-\alpha \pm \frac{1}{\sqrt{T}}$.
> All above guarantees hold deterministically, and simultaneously for all $T$, by Proposition 2.
>
> One may see the coverage gap $(h(t)+1)/T$ and think about setting $h(t)$ to $0$ or to set it as small as possible. The problem is that this leads to wild oscillations in the sets, like having too high a learning rate.
>
> In other words, the function $r(t)$ essentially acts as an adaptive learning rate, where the learning rate gets larger if the historical coverage gap is bigger. We show this formally using a Taylor expansion. Larger learning rates mean sharper discontinuities between adjacent sets — an extreme case is ACI with a high learning rate, which oscillates between infinite-sized and zero-sized sets. The benefit of setting, say, $h(t) = \sqrt{T}$, is that the effective learning rate is lower, which means smaller oscillations in the sets while still maintaining the desired coverage guarantees.
>
> > In the context of time series conformal prediction, how does this new method compare with other popular methods in terms of robustness, and computational efficiency?
>
> We can add a new subsection, titled “Comparison to Existing Methods”. We also evaluated against NexCP in the appendix. Please see the main response comment (5).
>
> > Despite demonstrating strong results across various experiments, the authors acknowledge potential limitations, including the challenge of proving asymptotic coverage, potential inapplicability to different time series, and limitations in scorecasting. The methods they develop are specific to cases where response is revealed at each time point. However, there are many settings in which we receive the response in a delayed fashion or in large batches.
>
> The COVID-19 experiment already showcases a setting with delayed feedback — the observations have a massive 4-week delay. This case is already covered by our framework. With a little bit more work, we have been able to significantly improve the COVID-19 results since the initial submission.
>
> Our results would be easily extended to the batched-sequential setting as well. This is a straightforward extension of our existing analysis. We can add a note about this fact in a new extensions section; see the main response comment.

---

> > ### Comment · Reviewer_aXqf · 2023-08-21
> >
> > Thank you for your reply. I've gone through the authors' rebuttal and am happy with the response. I have increased my score.

---

### Official Review · Reviewer_Cw13 · 2023-07-11

**Soundness:** 3 good
**Presentation:** 3 good
**Contribution:** 2 fair
**Rating:** 5
**Confidence:** 4

**Summary:**

This paper aims to provide asymptotically valid conformal prediction (CP) regions for the time series prediction problem. Similar to the adaptive conformal inference (ACI) framework [11], the main idea is to choose the appropriate quantile of the non-conformity scores (or equivalently tuning the miscoverage rate $\alpha$) at each time step to fulfill the required error rate. The framework performs an online quantile update while taking into account the sum of coverage errors overtime to stabilize the coverage better. They provided theoretical analysis that guarantees the asymptotic coverage of the predictions made by their framework. Numerical experiments on real datasets show that their method brings more stable and efficient prediction sets than ACI.

**Strengths:**

In general, I liked the idea of designing a PID-like framework. Thanks to this structure, the prediction regions turn out to be more stable and efficient. Moreover, it ensures that their framework never outputs infinite size prediction sets, which is a big advantage over ACI. Also, they provided many numerical experiments in the paper and the Appendix to illustrate the power of their proposed framework highlighting the effect of each block and different choices of hyperparameters.

**Weaknesses:**

1. The related work section is not well-written in general. Also, it does not represent the works in the intersection of conformal prediction and time series prediction in a fair manner. Indeed many good papers are missing, such as
	- Foygel Barber, R., Candes, E. J., Ramdas, A., and Tibshirani, R. J. Conformal prediction beyond exchangeability. arXiv preprint arXiv:2202.13415, 2022,
	- Sun, S. and Yu, R. Copula conformal prediction for multi-step time series forecasting. ArXiv, abs/2212.03281, 2022,
to mention a few.

2. The comparisons are limited to ACI. Though it is a natural baseline to compare with, many other CP frameworks are designed for the time series prediction problem. It is interesting to see how they behave compared to the methods proposed in this paper.

3. The scorecasting block seems unnecessary and, to some extent, redundant. Accounting for a wrong choice of learning model (e.g., a model that does not account for the seasonality effect) can not be a part of the design of a CP framework. This point is also mentioned in the paper in lines 286-288. However, it seems that the authors tried to put this block to force the similarity of their framework and the (well-known) PID controller.

4. There are numerous design choices (such as the saturation function) and hyperparameters in the framework, and it's worth noting that several of them may not be immediately intuitive for people in the Machine Learning or Conformal Prediction community when it comes to making appropriate choices.

5. Generally speaking, I am still unsure to what extent it is worth exploring frameworks with an asymptotic coverage property, while the popularity of conformal prediction lies in its ability to provide a finite-sample guarantee.

6. Minor Comments:
	- There are some typos in Equation 5.

	- Line115: There is a typo in the definition of the pinball loss (\tau should be replaced with \alpha).

	- Line 178: The equation reference should be (10) instead of (11).

	- In the proof of Proposition 2., the capital C has to be replaced with the small c.

	- Bringing the extensions of the proposed method (e.g., conformal risk control and Proposition 3. or conditional coverage) into the discussion section is unappealing.
	- In order to facilitate the comparisons for the readers,  it is a good idea to report the Avg. size of the prediction sets and Avg. coverage in the figures like what you have in the Appendix figures.

**Questions:**

Isn't it a good idea to provide a concrete explanation of why the proposed framework never returns infinite sets while this can happen in ACI?

**Limitations:**

The authors fairly mention the limitations of the proposed framework in the discussion section.

---

> ### Author Rebuttal · Authors · 2023-08-10
>
> Thank you for the accurate summary and the thoughtful review. We enjoyed engaging with your comments.
>
> The main response (in a separate comment) contains a summary of the extensive improvements we have made to the paper, including new experiments, finite-sample theoretical statements, automatic parameter selections, and a much-expanded section on related work. Herein we respond to the reviewer's specific concerns, and sometimes reference the main response.
>
> >The related work section is not well-written in general. Also, it does not represent the works in the intersection of conformal prediction and time series prediction in a fair manner. Indeed many good papers are missing, such as _Conformal prediction beyond exchangeability_ and _Copula conformal prediction for multi-step time series forecasting_, to mention a few. The comparisons are limited to ACI. Though it is a natural baseline to compare with, many other CP frameworks are designed for the time series prediction problem. It is interesting to see how they behave compared to the methods proposed in this paper.
>
> Thank you. We have improved the related work section, which we now realize was perhaps written too hastily. We added the papers you suggested as well as:
>
> “Exact and Robust Conformal Inference Methods for Predictive Machine Learning with Dependent Data” — Chernozhukov et al.
> “Conformal prediction with temporal quantile adjustments” — Lin et al.
> “Inductive conformal martingales for change-point detection” — Volkhonskiy et al.
>
> We are happy to include further references at the reviewer’s suggestion.
>
> Further, if you would find it valuable, we can add a new subsection titled “Comparisons to Existing Methods”, that contextualizes our work among the existing conformal time series papers. We also point out a comparison to NexCP in our original submission. Please see the main response (5) for more discussion about this.
>
> > The scorecasting block seems unnecessary and, to some extent, redundant. Accounting for a wrong choice of learning model (e.g., a model that does not account for the seasonality effect) can not be a part of the design of a CP framework. This point is also mentioned in the paper in lines 286-288. However, it seems that the authors tried to put this block to force the similarity of their framework and the (well-known) PID controller.
>
> It was not our intention to force a similarity to PID; rather, we believe that scorecasting can be useful in various practical situations where the base forecaster cannot easily be retrained, or cannot be easily updated, whether for computational or other practical reasons that have to do with the complexity of the original prediction pipeline. The COVID forecasting example is one such real-data situation. We have continued to work on this example and now adding a custom scorecaster makes a considerable improvement in the final coverage result — even though a massive, multi-institution effort was put into the initial model. This should clarify the real-world need.
>
> We have added text to explain better when scorecasting might be useful (and when it is not needed and may even hurt). Our initial version did not explain this sufficiently well, and we thank you for the critical feedback.
>
> There are numerous design choices (such as the saturation function) and hyperparameters in the framework, and it's worth noting that several of them may not be immediately intuitive for people in the Machine Learning or Conformal Prediction community when it comes to making appropriate choices.
>
> Agreed. In the new version of the paper, we provide default choices for all these parameters; see the main response (3).
>
> > Generally speaking, I am still unsure to what extent it is worth exploring frameworks with an asymptotic coverage property, while the popularity of conformal prediction lies in its ability to provide a finite-sample guarantee.
>
> Thank you. In the new version of the paper, we state finite-sample guarantees for all these algorithms; see (2) in the main response.
>
> > In order to facilitate the comparisons for the readers, it is a good idea to report the Avg. size of the prediction sets and Avg. coverage in the figures like what you have in the Appendix figures.
>
> For every plot, we have included a table that includes seven numerical comparisons:
>
> * Coverage.
> * Length of the longest sequence of errors.
> * Average set size.
> * 50%, 75%, 90%, and 95% quantiles of set size.
>
> We included these results for four different forecasting algorithms (ARIMA, FB Prophet, Theta model (theta = 2), and a Transformer).
>
> This addition is a small piece of a much larger effort we made to improve our experiments: see (1) of our main response.
>
> > Bringing the extensions of the proposed method (e.g., conformal risk control and Proposition 3. or conditional coverage) into the discussion section is unappealing.
>
> We have shortened the discussion section which was previously too long and distracting. We can add a separate subsection on extensions to the camera-ready version if that is deemed best.
>
> > Minor Comments
>
> Thank you for the fixes.

---

> > ### Comment · Reviewer_Cw13 · 2023-08-19
> >
> > Thanks for your detailed response to my review. You've addressed most of my concerns. I also like the idea of adding "Comparisons to Existing Methods."

---

### Author Rebuttal · Authors · 2023-08-10

We are grateful to the four expert and engaged reviewers, who took a clear interest in the paper and suggesting ways to improve it. Thank you!

All four said the practical benefits are substantial, as evidenced by comprehensive experiments. Three lean towards acceptance.

We hope to respond to the critical comments through the following extensive revisions:

**1. Major experimental revamp (see PDF attached)** We tripled the number of experiments and significantly extended our evaluations.

The COVID-19 experiment was completely reworked to look at state-level forecasts, and includes a custom scorecaster that learns geographic correlations between cases and deaths. This provides a large improvement over the CDC model used in official communications on COVID forecasting. See PDF Fig. 1.

We also included experiments with all combinations of the following.

_Forecasters:_ Transformer, Facebook Prophet, Theta(2), and AR(3).

_Datasets:_ Electricity demand in New South Wales, Returns of AMZN, MSFT, and GOOGL, and Temperature fluctuations in Delhi.

Each experiment now includes a table with these metrics:
* Coverage over the sequence.
* Longest continuous sequence of miscoverages.
* Mean set size.
* 50%, 75%, 90%, and 95% quantiles of set size.
(See PDF Figs 5 and 6; we couldn't include all tables due to space, and had to abbreviate/shrink the text. Apologies for this.)
All experiments now use score functions that allow for asymmetric prediction sets. The results are much improved. (See PDF Figs 2-4.)

**2. All results have been rewritten in finite samples.**
Reviewers noted that asymptotic coverage is less interesting than finite-sample coverage.

We rewrote the paper so _all results are finite-sample_, and give both upper- and lower- bounds on the coverage gap (the distance of the empirical coverage from alpha).

The bound in Proposition 1 is

$\frac{1}{T}\sum\limits_{t=1}^T\mathrm{err}_t=\alpha \pm \frac{b+\eta}{\eta T}$

And Proposition 2 is

$\frac{1}{T} \sum\limits_{t=1}^T\mathrm{err}_t=\alpha \pm \frac{ch(t)+1}{\eta T}$

Note: The bounds suggest that large $\eta$ is desirable from the perspective of coverage. However, when $\eta$ is massive, the gradient step $q_t + \eta g$ can over/under-shoot $s_{t+1}$, leading to sets that do not properly track the score quantile and instead oscillate.  See point (3) in this comment for an automated choice of $\eta$ that works well in all our experiments.

These propositions are not new content; the proofs are the same as the old ones. Previously, we stated the asymptotic version for ease of exposition, but the finite sample results were already proven.

Towards conditional coverage, Prop. 3 can be extended to provide simultaneous coverage over any covariate-dependent kernel-weighted time-windows. We can add this if reviewers would find it valuable.


**3. Automatic selection of the learning rate.**
We provided an automatic choice of the learning rate and default parameters for the integrator. This learning rate has good performance on all our datasets, which are relatively diverse. We recommend it for practical use. This avoids all hand-tuning. The setting is:

$\eta = 0.1 \hat{B}_t$,

Where $\hat{B}_t$ is an estimate of the max score computed within a trailing window. (Because this learning rate is technically changing online, it gains its validity via the integration argument, not the standard quantile tracking one.)

_All figures in the PDF use this learning rate._

The procedure is less sensitive to the integration parameters. For these, we provide defaults in the Appendix.

**4. New subsection on extensions**
Reviewers asked for a separate “Extensions” subsection in the camera-ready version, which we can add.

**5. More precise related work, and comparisons to SPCI.**

We propose a new subsection titled “Comparisons to Existing Methods”, if the reviewers would find it valuable. Its purpose would be to draw connections between our paper and related methods. Note that at the request of a reviewer, we already added a more comprehensive lit review of conformal prediction for time series.

We included evaluations against NexCP in our original appendix, but forgot to mention it in text. The “Trail” column is NexCP with a trailing window of weights. ACI typically performs better than NexCP, so we didn’t feel it was worth putting into the main text.

Regarding SPCI, our method is _valid in greater generality_ than SPCI. Importantly, SPCI relies on the following assumptions for validity:

* The use of the quantile random forest (QRF) algorithm.
* Both model assumptions, such as consistency, and distributional assumptions, such as smoothness.
* The result is intrinsically asymptotic due to the consistency requirement.

Thus, we believe that a direct comparison of SPCI and the method herein is not appropriate. The validity of SPCI is significantly weaker, perhaps along the lines of a standard nonparametric analysis. In contrast, our method has finite-sample validity averaged over time for any input sequence. However, it is possible to use the QRF/SPCI algorithm as the scorecaster, which would endow it with a distribution-free guarantee.

In our original submission, we did not pursue the QRF/SPCI angle because, in our initial experiments, the QRF was not a particularly effective time series forecasting method. It was outperformed by quantile LASSO (which we use for the COVID experiments) and the Theta model. The performance of the QRF tends to be “blocky” and high-variance; for an example, see plots on the authors’ GitHub: https://github.com/hamrel-cxu/SPCI-code/blob/main/tutorial_electric_EnbPI_SPCI.ipynb

If, after reading our response, the reviewers would still like to see an explicit comparison to SPCI, they can inform us, and we will include one (this essentially means adding QRF as a scorecaster, which is easy to do).

Finally, we once more thank the reviewers for their comments. The paper has improved a lot because of their feedback.

---

### Decision · Program_Chairs · 2023-09-21

**Decision:**

Accept (poster)

**Comment:**

This paper has been well-received by the reviewers, who all agree that the authors make a substantial contribution to uncertainty quantification for time series forecasting. A couple of issues have nevertheless been raised in the reviews, but these could be clarified in the rebuttal. Eventually, all reviewers are in favour of accepting the paper.